# FORMATION OF REPRESENTATIONS IN NEURAL NETWORKS

**Liu Ziyin**[1,3]**, Isaac Chuang**[1]**, Tomer Galanti**[2]**, Tomaso Poggio**[1]
[1]*Massachusetts Institute of Technology*
[2]*Texas A&M University*
[3]*NTT Research*

## ABSTRACT

Understanding neural representations will help open the black box of neural networks and advance our scientific understanding of modern AI systems. However, how complex, structured, and transferable representations emerge in modern neural networks has remained a mystery. Building on previous results, we propose the Canonical Representation Hypothesis (CRH), which posits a set of six alignment relations to universally govern the formation of representations in most hidden layers of a neural network. Under the CRH, the latent representations (R), weights (W), and neuron gradients (G) become mutually aligned during training. This alignment implies that neural networks naturally learn compact representations, where neurons and weights are invariant to task-irrelevant transformations. We then show that the breaking of CRH leads to the emergence of reciprocal power-law relations between R, W, and G, which we refer to as the Polynomial Alignment Hypothesis (PAH). We present a minimal-assumption theory proving that the balance between gradient noise and regularization is crucial for the emergence of the canonical representation. The CRH and PAH lead to an exciting possibility of unifying major key deep learning phenomena, including neural collapse and the neural feature ansatz, in a single framework.

## 1 INTRODUCTION

The success of deep learning is often attributed to its ability to learn meaningful latent representations from data (Bengio et al., 2013). These latent representations, progressively formed as data passes through the network's layers, are found to encode increasingly abstract features of the input:

$$x \to h^1 \to h^2 \to ... \to h^D \to \hat{y}, \tag{1}$$

where $x$ is the input, $\hat{y}$ the output, $D$ the network depth, and $h^i$ the activation of the $i$-th layer. For neural networks to perform well, the transformations between layers must capture meaningful structures in the data. Understanding how these latent representations are formed and structured is a foundational problem in deep learning, with implications for both theoretical understanding and practical applications. Despite significant advances, how neural networks organize and transform these internal representations remains an open question. This gap in understanding hinders our ability to design more efficient, interpretable, and generalizable models.

In this work, we seek to bridge this gap by introducing the Canonical Representation Hypothesis (CRH). At its core, the CRH posits that neural networks, during training, inherently align their representations with the gradients and weights. The satisfaction and breaking of a subset of CRH equations are found to delineate the universal phases, which are empirically observable scaling relationships between the weights, activations, and gradients. The CRH reveals a striking aspect of representation learning: there may exist a set of *universal equations* that govern the formation of representations and *universal phases* which distinguish the layers in modern neural networks, independent of the task, architecture, or loss function. Thus, the CRH provides an useful perspective on how neural networks evolve toward compact and interpretable solutions.

Our main contributions are the following:

1. Proposal and justification of the CRH which states that within a layer, the neuron gradients, latent representation, and parameters are driven into mutual alignment after training, due to noise-regularization balance (Section 3 and 4);

2. Identification of mechanisms that break alignments, quantified via a Polynomial Alignment Hypothesis, which predicts power law scaling behaviors that characterize distinct phases of neural networks arising when the CRH is broken (Section 5).

The experiments are presented in Section 6. Section 7 discusses the implications of CRH to the formation of neural representations and the connection of the CRH to prior observations. Related works are discussed in Section 2. The proofs are left to the Appendix Section B.

## 2 RELATED WORKS

Empirical results show that the representations of well-trained neural networks share universal characteristics (Maheswaranathan et al., 2019; Huh et al., 2024; Ziyin et al., 2025). For us, a closely related phenomenon is the neural collapse (NC) (Papyan et al., 2020), which studies how structured and noise-robust low-rank representations emerge in a classification model. The CRH can be seen as a generalization of NC because one can prove that when restricted to certain settings, the CRH is equivalent to the NC (Section 7). Another related phenomenon is the neural feature ansatz (NFA) (Radhakrishnan et al., 2023), which shows that the weight matrices of fully connected layers evolve according to the gradient outer product during training. However, the NFA studies the weight evolution, not the representations. Empirical power-laws are known to exist in large neural networks (Kaplan et al., 2020; Bahri et al., 2024), which relates the model performance to their sizes. The power laws discovered in our work are different because they are reciprocal relations that relate dual objects (R, G, W) to each other rather than to the performance. Other related works are discussed in the context where they become relevant. More related works are discussed in Appendix A.

## 3 CANONICAL REPRESENTATION HYPOTHESIS

Let us consider an arbitrary hidden layer $h_b$ of any model after a linear transformation:

$$h_b = W h_a(x). \tag{2}$$

For convention, $h_b$ is called the "preactivation" of the next layer, and $h_a$ is the "postactivation" of the previous layer. The gradients of the representations are also of interest: $g_a = -\nabla_{h_a}\ell$ and $g_b = -\nabla_{h_b}\ell$, where $\ell$ is the sample-wise loss function. We note the generality of this setting, as $h_a$ can be an arbitrarily nonlinear function of $x$, and $f(x) = f(h_b(x))$ can be another arbitrary transformation. Also, letting $h_a = (h'_a(x), 1)$ accounts for when there is a trainable bias.[1]

Much recent literature has suggested that the quantities $h_a$, $h_b$, $W$ and their gradients are correlated with each other after or throughout training. The neural collapse phenomenon suggests that in a deep overparameterized classifier, $\mathbb{E}[h_a h_a^\top] \propto W^\top W$ for the penultimate fully connected layer (Papyan et al., 2020; Xu et al., 2023b; Ji et al., 2021; Kothapalli, 2022; Rangamani et al., 2023). In the study of kernel and feature learning, a primary mechanism of how the neural tangent kernel changes is that after a few steps of update, the representations become correlated with the weights (Everett et al., 2024) and so the quantity $W h_a$ will be significantly away from zero, which also implies a strong relationship between $W^\top W$ and $\mathbb{E}[h_a h_a^\top]$. The recent work on neural feature ansatz shows that $W^\top W \propto \nabla_{h_a} f \nabla_{h_a}^\top f$ for fully connected networks (Radhakrishnan et al., 2023; Gan & Poggio, 2024). The idea that the latent variables will become correlated with the weight updates is also a central notion in the feature learning literature (Yang & Hu, 2020; Everett et al., 2024).

Taken together, these results suggest a simple and unifying set of equations that can describe a fully connected layer after (and perhaps during) training. Let $c \in \{a, b\}$ and define $H_c = \mathbb{E}[h_c h_c^\top]$, $G_c = \mathbb{E}[g_c g_c^\top]$, and $Z_c = M_c M_c^\top$, where $M_a = W^\top = M_b^\top$. One can imagine six alignment relations between all the quantities within the same layer:

$$\text{representation-gradient alignment (RGA): } H_c \propto G_c, \tag{3}$$

$$\text{representation-weight alignment (RWA): } H_c \propto Z_c, \tag{4}$$

$$\text{gradient-weight alignment (GWA): } G_c \propto Z_c, \tag{5}$$

where $\mathbb{E}$ denotes the averaging over the training set. Because $h_b$ comes after $h_a$ during computation, we will refer to the alignment between any of the $b$-subscript matrices as a *forward* relation and all $a$-subscript matrices as a *backward* relation.

---

[1] As an example, consider a two-layer network, $f(x) = W_2 \sigma(W_1 x)$, with fully connected layers. For the first layer, $h_a^1 = x$, and $h_b^1 = W_1 x$. For the second layer, $h_a^2 = \sigma(W_1 x)$ and $h_b^2 = W_2 h_a^2$.

For the formation of representations in neural networks, the most important relations in Eq. (3)-(5) are perhaps the forward and backward RGA, as they directly relate the representations $\mathbb{E}[hh^\top]$ to their gradients with respect to the loss function. Because there is no scientific reason for us to believe that the forward representation is more important than the backward representation or vice versa, one should study both directions carefully. The backward RWA in its general form has not been discussed in the literature but has been implicitly studied in the particular setting of neural collapse, which happens in the penultimate layer of an image classification task (Section 7), while the backward GWA seems to be unknown to the best of our knowledge. The backward GWA is not identical to the neural feature ansatz (NFA) but can be seen as an equivariant correction to the NFA (Section 7), and the forward GWA is also unknown to the community. Adding together the forward and backward versions, there are six alignment relations. That these relations simultaneously hold for any fully connected layer will be referred to as the **canonical representation hypothesis** (CRH).

Three scientifically fundamental questions are thus (1) does there exists a rigorous set of assumptions under which the CRH can be proved; (2) what mechanisms can cause the CRH to break; and (3) can predictions of the CRH and its breaking be empirically observed in realistic deep neural network settings? We devote the rest of the paper to answering these three questions in the given order, and then we collect insights from all these answers.

## 4 NOISE-REGULARIZATION BALANCE LEADS TO ALIGNMENT

**Notation** $\Delta A$ denotes the difference in the quantity $A(\theta)$ after one step of training algorihtm iteration at time step $t$: $\Delta A := A(\theta_{t+1}) - A(\theta_t)$. $\eta$ denotes the learning rate and $\gamma$ denotes the weight decay. $\mathbb{E}$ denotes the empirical average over the training set. $\ell(x, \theta)$ denotes the per-sample loss function, where $x$ is the data point and $\theta$ is the parameters. Its empirical average is the empirical loss $L$: $L(\theta) = \mathbb{E}[\ell(x, \theta)]$.

In this section, we present a formal and rigorous framework under which the CRH can be proved. As will become clear in the next section, this proof also explains how and when the CRH may fail. The problem setting is the same as in Eq. (2). The training proceeds in an online learning setting in which the training proceeds with weight decay of strength $\gamma$. We make the following assumption.

**Assumption 1** (Mean-field norms). *The norms of $g$ and $h$ approximate their empirical averages:* **(A1)** $\|h_a\|^2 = \mathbb{E}[\|h_a\|^2]$, **(A2)** $\|g_b\|^2 = \mathbb{E}[\|g_b\|^2]$.

This assumption holds, for example, for a high-dimensional Gaussian random vector, whose norm is of order $O(d)$ with a $\sqrt{d}$ standard deviation. A1 also holds automatically if the representations are normalized. Note that only a subset of all the assumptions is needed to prove each equation we derive below. For example, Eq. (6) below only requires A1 to prove. The minimal set of assumptions required to prove each equation in this section are stated in Section B.3. We discuss the main intuition for the proof in the main text, and present the formal theorem at the end of the section.

**Forward alignment.** Consider the time evolution of $h_b h_b^\top$ during SGD training:

$$\Delta(h_b(x)h_b^\top(x)) = \eta(\|h_a\|^2 g_b h_b^\top + \|h_a\|^2 h_b g_b^\top - 2\gamma h_b h_b^\top) + \eta^2 \|h_a\|^4 g_b g_b^\top + O(\eta^2 \gamma + \|\Delta(h_a h_a^\top)\|),$$

where $\eta$ is the learning rate. At the end of training, the learned representations should reach stationarity and so $\Delta\mathbb{E}[h_b h_b^\top] \approx 0$.[2] Taking the expectation of both sides, we obtain

$$0 = \underbrace{z_b \mathbb{E}[g_b h_b^\top] + z_b \mathbb{E}[h_b g_b^\top]}_{\text{learning}} - \underbrace{2\gamma \mathbb{E}[h_b h_b^\top]}_{\text{regularization}} + \underbrace{\eta z_b^2 \mathbb{E}[g_b g_b^\top]}_{\text{noise}}, \tag{6}$$

where $z_b = \mathbb{E}[\|h_a\|^2]$. The noise term is due to the discretization error of SGD and can be significant either when the step size is large, or the gradient is noisy. The mechanism behind this alignment is that while the gradient noise expands the representation, weight decay contracts it. When the dynamics reaches approximate stationarity, the dynamics due to learning plus these two effects must balance at the end of training.

Now, if additionally either (a) $\mathbb{E}[\Delta W] = 0$ (namely, at a local minimum) or (b) $\mathbb{E}[\Delta(WW^\top)] = 0$ holds (Section B), the weight will also align with the cross terms between $g_b$ and $h_b$: $WW^\top \propto$

---

[2]See Figure 21 for the evolution of $\Delta H$.

$\mathbb{E}[g_b h_b^\top] + \mathbb{E}[h_b g_b^\top]$, which leads to the effect that the learning term above must also balance with the regularization term. Thus, eventually, the regularization effect will have to be balanced with the gradient noise. Lastly, if both (a) and (b) hold, the alignment between all three matrices emerges: $G_b \propto H_b \propto Z_b$.

**Backward Alignment.** One can similarly derive condition for $\mathbb{E}[\Delta G_a] = 0$: $z_a(\mathbb{E}[h_a g_a^\top] + \mathbb{E}[g_a h_a^\top]) + \eta z_a^2 \mathbb{E}[h_a h_a^\top] = 2\gamma \mathbb{E}[g_a g_a^\top]$, where we have defined $z_a = \mathbb{E}[\|g_b\|^2]$. The forward CRH can then be derived if $W^\top W$ and $W$ reaches stationarity.

The following theorem formalizes these results.

**Theorem 1.** *Under Assumption 1, when* $\mathbb{E}[\Delta H_a] = 0$, $\mathbb{E}[\Delta G_b] = 0$, $\mathbb{E}[\Delta(WW^\top)] = 0$, *and* $\mathbb{E}[\Delta(W^\top W)] = 0$, *there exist real-valued constants* $c_1$, $c_2$, $c_3$, $c_4 > 0$ *such that*

$$WW^\top + c_1 \mathbb{E}[g_b g_b^\top] = c_2 \mathbb{E}[h_b h_b^\top], \quad W^\top W + c_3 \mathbb{E}[h_a h_a^\top] = c_4 \mathbb{E}[g_a g_a^\top]. \tag{7}$$

*Additionally, if at a local minimum,*

$$WW^\top \propto \mathbb{E}[g_b g_b^\top] \propto \mathbb{E}[h_b h_b^\top], \quad W^\top W \propto \mathbb{E}[h_a h_a^\top] \propto \mathbb{E}[g_a g_a^\top]. \tag{8}$$

**Remark.** *A strength of this derivation is that it is oblivious to the loss function, the model architecture, or the type of activation used (as long as the second moments exist). This may explain the wide applicability of the RGA observed in Section 6. The above alignment relations can be seen as a type of fluctuation-dissipation theorem in theoretical physics (Kubo, 1966), which states that in a driven stochastic dynamics, the fluctuation of the force must balance with the rate of energy loss – a fundamental law first discovered by Einstein (1905). Prior applications of the fluctuation-dissipation theorem to deep learning have focused on the covariance of the model parameters (Yaida, 2018; Liu et al., 2021) and are not directly relevant to the representations.*

## 5 CRH BREAKING AND POLYNOMIAL ALIGNMENT HYPOTHESIS

While the CRH can be found to hold for many scenarios, it is highly unlikely that it always holds perfectly and for every layer (e.g., see Section 6). In this section, we study what happens if the CRH is broken; we then suggest two mechanisms which cause the CRH to break. The following theorem shows that all six relations are intimately connected, even if only a subset of the CRH holds. For a square matrix $A$, we use $A^{-n} := (A^+)^n$ to denote the $n$−th power of the pseudo inverse of $A$, and $A^0 = AA^+$ is an orthogonal projection matrix to the column space of $A$.

**Theorem 2** (CRH Master Theorem). *Let $A$, $B$, $C$ be a permutation of $\mathbb{E}[hh^\top]$, $\mathbb{E}[gg^\top]$, and $Z$, and let $\tilde{D} := PDP$ be a projected version of $D$ for a projection matrix $P$. Then,*

1. *(Directional Redundancy) if any two forward (backward) alignments hold, all forward (backward) alignments hold;*
2. *(Reciprocal Polynomial Alignments) if one of any forward alignments and one of any backward alignments hold, there exists scalars $\alpha_c$, $\beta_c$, and $\delta_c$ satisfying $-1 \le \alpha_c, \beta_c, \delta_c \le 3$ such that*

$$\tilde{A}_c^{\alpha_c} \propto \tilde{B}_c^{\beta_c} \propto \tilde{C}_c^{\delta_c}, \tag{9}$$

   *(as detailed in Table 1) where $c \in \{a, b\}$ denotes the backward and forward relations respectively, and the corresponding projection $P_c \in \{Z_c^0, \mathbb{E}[h_c h_c^\top]^0, \mathbb{E}[g_c g_c^\top]^0\}$, e.g. such that $\tilde{A} = P_c A P_c$.*
3. *(Canonical Alignment I) If (any) one more relation holds in addition to part 2, then all six alignments hold in the $Z^0$ subspace; in addition, at a local minimum, all six alignments hold;*
4. *(Canonical Alignment II) If all six alignments hold, $\mathbb{E}[hh^\top] \propto \mathbb{E}[gg^\top] \propto Z \propto P$, where $P$ is an orthogonal projection matrix.*

The idea behind the proof is that there is some redundancy in the six matrices: every forward relation implies a backward relation and vice versa. As an example, if $Z_a \propto H_a$, then we also have $Z_b^2 \propto H_b$, which can be obtained by multiplying $W$ on the left and $W^\top$ on the right.

Part (4) of the theorem clarifies what it means to satisfy the CRH: the latent representation is fully compact, where weight and representation are only nonvanishing in the subspaces where the gradient is nonvanishing. Moreover, the weight matrix does nothing but rotates the representation, implying that the information processing is invertible once the CRH is fully satisfied. This is consistent with the observation that once a layer has an almost perfect alignment, all the layers after it also

have perfect alignment, a sign that the representation cannot be further compressed (Section 6). Therefore, the CRH is consistent with the observation that last layers of neural networks are low-rank and invariant to irrelevant features.

Part (2) is especially relevant when the CRH is broken. Depending on which subset of the hypotheses holds, the learning process may be classifiable into as many as $2^6 = 64$ phases. In different phases, the learning dynamics and the found solution will likely be different due to different scaling relations. For example, positive exponents between $Z_a$ and $H_a$ will imply that the layer is enhancing the principle components of $H_a$, while suppressing the lesser features; a negative exponent would imply the converse. Even if we remove the redundancy implied by the theorem, there are still at least 22 phases. One can also define additional phases according to the ordering of the degree of breaking for each relation, which gives $6! = 720$ phases, although these ordering phases may not have a major influence. In many experiments, we performed, at least one of the forward relations and one of the backward relations are observed to hold very well (e.g., see Figure 8). This means that one is quite likely to observe the power-law relations predicted in Table 1. We also find it common for different layers to be in different phases, even within the same network. We discuss more potential meanings and examples in Section D.

Broadly interpreted, part (2) predicts a power law relation between the spectrum of all six matrices, which is also what we observe in almost all experiments (Section 6). What is quite surprising is that almost all positive exponents we observed are within the range $[1/3, 3]$, which is exactly the range of exponents that the theorem predicts (e.g., see Section C.8). Formally, that the $H$, $Z$, and $G$ are polynomially related can be called the "**Polynomial Alignment Hypothesis (PAH)**" and is a natural extension of the CRH. That scaling relations can be used to characterize different phases of matter is an old idea in science. In physics, phases can be classified according to their scaling exponents, and having a different set of exponents implies that the underlying dynamics and mechanism are entirely different (Pelissetto & Vicari, 2002). This connection corroborates our physics-inspired proof.

**Breaking of CRH.** A major remaining question is whether we can find mechanisms such that the CRH breaks. The theory in the previous section implies one primary mechanism that the CRH breaks. For all six alignments to hold, it needs to be the case that both $\mathbb{E}[\Delta W]$, $\mathbb{E}[\Delta Z]$ are zero, but these two conditions may not be easily compatible with each other, as they together imply that $\Delta W$ has zero variance. While this may be possible for some subspaces (e.g., see Ziyin et al. (2024a)), it does not hold for every subspace. In fact, it is easy to show that unless the minibatch size is large enough the SGD updates will never reach zero variance that is $\Delta W \neq 0$ even for $t \to \infty$ (see Lemma 8 in Xu et al. (2023b); see also Xu et al. (2023a)). Thus, the competition between reaching a training loss of zero and the need to reach a stationary fluctuation is a primary cause of the CRH breakage.

| Phase | Back. Alignment | Forw. Alignment | Back. Power Law | Forw. Power Law | NC | NFA | CU | llm |
|---|---|---|---|---|---|---|---|---|
| CRH | $H_a \propto Z_a \propto G_a$ | $H_b \propto Z_b \propto G_b$ | - | - | ✓ | ✓ | ✓ | |
| back. CRH | $H_a \propto Z_a \propto G_a$ | - | - | $\tilde{H}_b^0 \propto \tilde{Z}_b^0 \propto \tilde{G}_b$ ($H_b \propto Z_b^2$) | ✓ | ✓ | | |
| forw. CRH | - | $H_b \propto Z_b \propto G_b$ | $\tilde{H}_a \propto \tilde{Z}_a^0 \propto \tilde{G}_a^0$ ($Z_a^2 \propto G_a$) | - | | | | ✓(3-6) |
| 1 | $H_a \propto G_a$ | $H_b \propto G_b$ | $\tilde{H}_a^0 \propto \tilde{Z}_a \propto \tilde{G}_a^0$ | $\tilde{H}_b^0 \propto \tilde{Z}_b \propto \tilde{G}_b^0$ | | | | ✓(3-6) |
| 2 | $H_a \propto Z_a$ | $H_b \propto Z_b$ | $\tilde{H}_a \propto \tilde{Z}_a \propto \tilde{G}_a^0$ | $\tilde{H}_b \propto \tilde{Z}_b \propto \tilde{G}_b^0$ | | | ✓ | |
| 3 | $G_a \propto Z_a$ | $G_b \propto Z_b$ | $\tilde{H}_a^0 \propto \tilde{Z}_a \propto \tilde{G}_a$ | $\tilde{H}_b^0 \propto \tilde{Z}_b \propto \tilde{G}_b$ | ✓ | | | |
| 4 | $H_a \propto G_a$ | $H_b \propto Z_b$ | $\tilde{H}_a \propto \tilde{Z}_a^0 \propto \tilde{G}_a$ | $\tilde{H}_b \propto \tilde{Z}_b \propto \tilde{G}_b^{-1}$ | | | | |
| 5 | $H_a \propto Z_a$ | $H_b \propto G_b$ | $\tilde{H}_a^3 \propto \tilde{Z}_a^3 \propto \tilde{G}_a$ | $\tilde{H}_b \propto \tilde{Z}^2 \propto \tilde{G}_b$ | | | ✓ | ✓(3-6) |
| 6 | $H_a \propto G_a$ | $G_b \propto Z_b$ | $\tilde{H}_a \propto \tilde{Z}_a^2 \propto \tilde{G}_a$ | $\tilde{H}_b \propto \tilde{Z}_b^3 \propto \tilde{G}_b^3$ | | | | ✓(1) |
| 7 | $G_a \propto Z_a$ | $H_b \propto G_b$ | $\tilde{H}_a^{-1} \propto \tilde{Z}_a \propto \tilde{G}_a$ | $\tilde{H}_b \propto \tilde{Z}_b^0 \propto \tilde{G}_b$ | ✓ | | | |
| 8 | $H_a \propto Z_a$ | $G_b \propto Z_b$ | $\tilde{H}_a^2 \propto \tilde{Z}_a^2 \propto \tilde{G}_a$ | $\tilde{H}_b \propto \tilde{Z}_b^2 \propto \tilde{G}_b^2$ | | | ✓ | ✓(2) |
| 9 | $G_a \propto Z_a$ | $H_b \propto Z_b$ | $\tilde{H}_a \propto \tilde{Z}_a^0 \propto \tilde{G}_a^0$ | $\tilde{H}_b^0 \propto \tilde{Z}_b^0 \propto \tilde{G}_b$ | ✓ | | | |

Table 1: The reciprocal polynomial relations of the CRH Master Theorem. When one forward relation and one backward relation hold simultaneously, all six matrices are polynomially aligned in a subspace (Theorem 2). Each scaling relationship can be regarded as a possible phase for the layer during actual training. The right panel shows how existing observations about neural networks fit into the phase diagram. A ✓ denotes that this phenomenon is compatible with the specified phase. NC refers to the neural collapse. NFA refers to the neural feature ansatz. CU (correlated update) refers to the (idealization of the) common observation that $h_a$ is correlated with $W$ a few steps after training (Everett et al., 2024). The llm column shows the compatibility of the scaling relation for transformer observed in Figure 3.

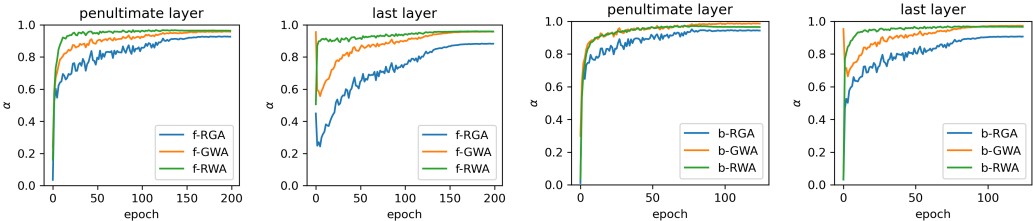

Figure 1: Six alignment relations in the penultimate layer and output layer of a ResNet18 trained on CIFAR-10 (**res1**). **Left**: forward CRH. **Right**: backward CRH. We see that all six relations hold significantly across two fully connected layers. Also, we show that the matrix $\mathrm{cov}(g, h)$ is well aligned with $WW^\top$ in the appendix Section C.7, which is a strong piece of evidence supporting the key theoretical step that the cross terms will be aligned with the weights (and $G, H$).

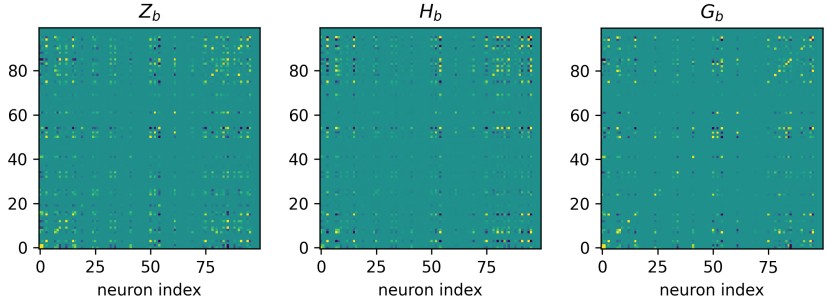

Figure 2: Penultimate layer of the conjugate matrices $(H, G, Z)$ after training (**fc2**). This is an example of CRH being well satisfied, where all three matrices are well aligned after training.

Because the rank tends to decrease for later layers in networks performing classification, we expect that CRH holds better for later layers than for earlier layers.

This problem is especially troublesome in the layers where $\mathbb{E}[\Delta W]$ has a high rank, which holds true for the earlier layers of the network but not the later layers (Xu et al., 2023a). This is consistent with the observation that the CRH holds much better in the latter layers than in the beginning layers. This analysis is consistent with the fact that a stronger alignment is strongly correlated with a more compact representation (Figure 4).

**Finite-Time Breaking of CRH.** To leading order in $\gamma$, the proof of the CRH implies that

$$G_b + O(\gamma) \propto \gamma^2 H_b \propto \gamma^2 WW^\top, \quad H_a + O(\gamma) \propto \gamma^2 G_a \propto \gamma^2 W^\top W. \tag{10}$$

This means that when $\gamma$ is small, the forward alignment between $H_b$ and $WW^\top$ are strong (because the prefactor is independent of $\gamma$), while the other two are weak – because the huge disparity between the two matrices, it might take gradient descent longer than practical to reach such a solution. Similarly, the backward alignment between $G_a$ and $W^\top W$ is strong for a small $\gamma$. This prediction will be verified in Section 7 when we discuss the neural feature ansatz.

## 6 EXPERIMENTS

In this section, we present experimental evidence that supports predictions resulting from the CRH. We also perform experiments to test mechanisms that break the CRH. When the CRH is broken, we put special emphasis on verifying the RGA, as it directly links to the formation of representations and may arguably be the most important relation of the three subsets.

**Metric for alignment.** We would like to measure how similar and well-aligned the six matrices in the CRH are. Let $A$ and $B$ be two square matrices, each with $d^2$ real-valued elements. We use the Pearson correlation to measure the alignment between two matrices (Herdin et al., 2005):

$$\alpha(A, B) := \frac{1}{K} \left( \frac{1}{d^2} \sum_{ij} A_{ij} B_{ij} - \frac{1}{d^4} \sum_{ij} A_{ij} \sum_{ij} B_{ij} \right), \tag{11}$$

where $K$ is a normalization factor that ensures $\alpha \in [-1, 1]$. $\alpha(A, B)$ will be referred to as the **alignment** between the two matrices $A$ and $B$. Note that $\alpha = \pm 1$ if and only if $A = c_0 B$ for some

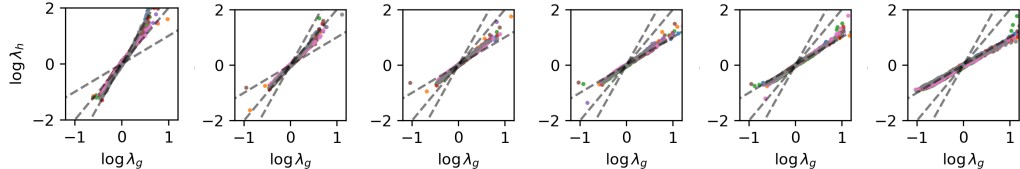

Figure 3: The power-law alignment between the eigenvalues $\lambda_h$ and $\lambda_g$ of $H_b$ and $G_b$ in a six-hidden layer transformer (**llm**). Left to Right: first to the penultimate layers. The grey dashed lines show the power-law relations $\lambda_h \propto \lambda_g^\alpha$ for $\alpha = 1, 2, 3$ respectively. We see that the first layer has an exponent of 3, the second has an exponent of 2, and all the layers after it are observed to have an exponent of 1. Different colors show different heads within the same layer. The range of the power exponents is in almost perfect agreement with the predicted range in Table 1. Referring to the table, this implies that these layers are in phases 5, 8, and 6, respectively. The setting is the same as the LLM experiment. Also, see Section C.8 for fully connected nets.

constant $c_0$. Therefore, $\alpha$ can be seen as a quantitative metric for the alignment and if $|\alpha| = 1$, the alignment is perfect. Our initial pilot experiments suggest that the alignment effects are the strongest when the $h$ and $g$ are normalized and if the mean of each is subtracted.[3] Thus, we always normalize $h$ and $g$ and subtract the mean to be consistent in the experiments. This is equivalent to measuring $\text{cov}(\hat{h}, \hat{h})$ and $\text{cov}(\hat{g}, \hat{g})$, where cov denotes covariance and $\hat{a} = a/\|a\|$. As a notational shorthand, we use $\alpha_{ab,cd}$ to denote $\alpha(\text{cov}(a, b), \text{cov}(c, d))$ for the rest of the paper.

**Settings.** We experiment with the following settings and name each setting with a unique identifier. **fc1**: Fully connected neural networks trained on a synthetic dataset that we generated using a two-layer teacher network. This experiment is used for a controlled study of the effect of different hyperparameters. **fc2**: the same as fc1, except that the output dimension is extended to 100 and the input distribution interpolates between an isotropic and nonisotropic distribution. **res1**: ResNet-18 (11M parameters) for the image classification; **res2**: ResNet-18 self-supervised learning tasks with the CIFAR-10/100 datasets. **llm**: a six-layer eight-head transformer (100M parameters) trained on the OpenWebText (OWT) dataset. The details of training methods are described in Section C.

**CRH.** We start with the supervised learning setting with ResNet-18 trained on CIFAR-10. We measure the covariances matrices with data points from the test set. Figure (3) shows that very good alignment $\alpha > 0.7$ is achieved quite early in the training, and continues to improve during the later stage of training. This case might remind some readers of neural collapse (NC) – because NC is also most significant in the penultimate layer of large image classifiers. As we will show in the next section, in the interpolation regime of classification tasks, the NC is equivalent to the CRH.

Another example of CRH is provided in Figure 8 below, in case of a fully connected network in a regression task. The result shows that when the weight decay is not too small, the CRH is quite close to being perfectly satisfied for intermediate layers. Examples of the representation and the dual matrices are presented in Figure 2 for a layer that (almost) satisfies the CRH.

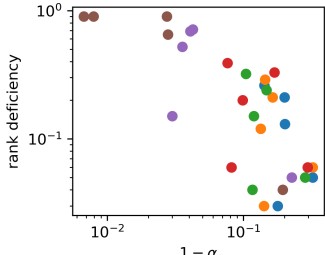

**Breaking of CRH.** A clear evidence for the correctness of the theory is that at a small weight decay, the strongest alignments are the forward-RWA and the backward-RGA, which will be presented in the experiment in Figure 8 after we discuss the relationship of the CRH to the NFA.

Two indirect evidences are that (1) the compactness of the representation is found to be negatively correlated with the alignment level (Figure 4), and (2) the observed positive exponents between the eigenvalues of the matrices are almost aways within the range $[1/3, 3]$, which is the predicted range by Theorem 2 (Figure 3).

Figure 4: The rank deficiency and the backward $\alpha_{gg,hh}$ in fully connected nets (**fc2**). The rank of representation is strongly negatively correlated with $\alpha$. Here, every color is a different weight decay (from $10^{-6}$ to $10^{-4}$), and every point is a different layer in the net. The setting is the same as the fully connected net experiment.

**RGA.** A relation of particular interest to the formation of representations is the RGA, which predicts that the represenations are aligned with the gradients across them. Now, we show that the

---

[3]These may be due to the fact that the gradient and activation have rare outliers that tend to disturb the balance. See Section C.2 for the RGA on transformers without subtracting the mean.

Figure 5: Alignment between $\text{cov}(h, h)$ and $\text{cov}(g, g)$ in a six-layer transformer trained on the OWT dataset (**llm**). From **left** to **right**: layer 1, 2, 4, 5. Also, see layer 3 in Figure 15. The shaded region shows the variation (min and max) across eight different heads in the same layer. The RGA is significnatly stronger than the alignment between initial and final representation, and the alignment between different heads.

RGA holds well across a broad range of tasks in common training settings. When the CRH holds, the RGA holds as well, and so the experiments in the CRH section already shows that the RGA holds for the last layers of ResNet.

*Large Language Models* (**llm**). We measure the covariances of the output of each attention head in every layer. See Figure 5 for the evolution of the alignment and Figure 14 for examples of the representation. Three baselines of comparisons are (1) the alignment between the covariances of a rank-40 (roughly equivalent to the actual ranks of $\text{cov}(h, h)$ and $\text{cov}(g, g)$) random projection of two 200 dimensional isotropic Gaussian, which stays close to $0.14$, (2) the alignment between the feature covariance of different heads, which starts high but drops to a value significantly lower than $\alpha_{gg,hh}$, (3) the alignment between the initial feature and current feature for the same head, which starts from 1 and also drops quickly. The RGA is found to hold stronger than the other baselines.

*Self-Supervised Learning* (**res2**). Self-supervised learning focuses on learning a good and versatile representation without knowing the labels for the problem. See Figure 6 for the time evolution of $\alpha$. We see that the RGA holds well for the fully connected layers, but not so strongly for the conv layers. Experiments do show increased $\alpha$ values if we decrease the batch size and increase $\gamma$. Also, see Figure 13 for examples of the representations in the convolutional layers. We see very good qualitative alignment between the two matrices.

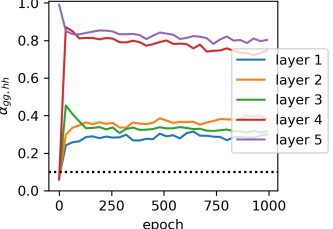

Figure 6: The alignment for different layers during 1000 training epochs (**res2**). Layers 1-3 are convolutional layers, and layers 4-5 are fully connected ones.

*Fully Connected Nets* (**fc1**). We also perform a systematic exploration of how hyperparameters of training and model architectures affect the formation of representations. Results presented in Section C.6 show the following phenomena: (1) *Gradual Alignment*: deeper layers tend to have better alignment; if a layer has an almost perfect alignment ($\alpha \approx 1$), then any layer after it will also have almost perfect alignment (note the similarity of this phenomenon with neural collapse); this is consistent with the tunnel effect discovered in (Masarczyk et al., 2024); (b) *Critical Depth and Alignment Separation*: batch size $B$ affects the alignment significantly; earlier layers have worse alignment as $B$ increases; later layers have better alignment as $B$ increases; wider networks have similar level of alignment as thinner ones for SGD; for Adam, it is more subtle: early layers have worse alignment for a large width, later layers have better alignment for a large width; (c) weight decay $\gamma$ affects the alignment significantly and systematically; larger $\gamma$ always leads to better alignment.

## 7 INSIGHTS

This section first studies the implication of the CRH (mainly the RGA) for the representation of neural networks (7.1-2). We then clarify the relation of the CRH to NC and NFA (7.3-4).

### 7.1 PLASTICITY-EXPRESSIVITY COUPLING

A direct interpretation of the RGA is that the plasticity of neurons in neural networks is strongly coupled to their expressivity *after training*:

$$\underbrace{\mathbb{E}[h_b h_b^\top]}_{\text{expressivity}} \propto \underbrace{\mathbb{E}[g_b g_b^\top]}_{\text{plasticity}}. \tag{12}$$

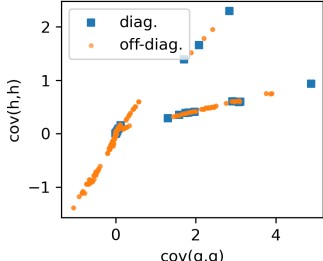

Figure 7: The response diversity (the diagonal terms of the covariance) and correlation is coupled to the plasticity of neurons (**fc1**).

The first term directly measures the variability of the neuron response to different inputs, thus measuring how expressive or capable this neuron is. The second term measures the degree of plasticity because the gradient for the weight matrix prior to $h_b$ is proportional to $g_b h_a^\top$, and if $g_b$ is zero with zero variance, the input weights to this neuron will never be updated. It is, therefore, a measure of plasticity. This coupling is quite unexpected because the expressivity should be independent of plasticity at initialization. See Figure 7 for a scattering plot of how the elements of $\mathbb{E}[hh^\top]$ are strongly aligned with that of $\mathbb{E}[gg^\top]$. While this experiment shows a strong alignment between $H$ and $G$, it also shows a limitation of our theory. If the activation norm $h$ is really independent of its direction $h/\|h\|$, one would only see one branch in the figure, whereas there are three branches. This can be another reason for CRH breaking.

## 7.2 Feature-Importance Alignment and Invariant Learning

Another direct interpretation of $\mathbb{E}[gg^\top]$ is the saliency of a latent feature. The quantity $\mathbb{E}[gg^\top]$ measures how the loss function changes as a neuron activation is changed by a small degree and is often used on the input layer to measure the importance of a feature (Selvaraju et al., 2017; Adadi & Berrada, 2018). This interpretation justifies many heuristics for understanding neural networks: take the principal components of a hidden layer and study its largest eigenvalues. If there is no direct link between the magnitude of the eigenvalues in these latent representations and their importance in affecting the learning (measured by the training loss), these analyses will be unreasonable.

Let $\hat{n}$ be any unit vector that encodes a feature irrelevant to the task. The following theorem shows that the invariance of the representation to such features is equivalent to the invariance of the model.

**Proposition 1.** *Let $f(h(x))$ be a model whose hidden states $h$ obey RGA. Let $\hat{n}$ be a vector and $\epsilon$ a scalar. The following statements are equivalent: (1) $\ell(f(h + \epsilon\hat{n})) = \ell(f(h)) + O(\epsilon^2)$; (2) $\mathbb{E}[hh^\top]\hat{n} = 0$; (if the CRH also holds) $W\hat{n} = 0$, and $G\hat{n} = 0$.*

In some sense, Proposition 1 can be seen as a generalization of NC (next section) because it applies to both regression and classification tasks. Note that there are two ways for the condition $\ell(f(h+\epsilon\hat{n})) = \ell(f(h))$ to be satisfied: (1) $f(h + \epsilon\hat{n}) = f(h)$, which means that when the model output itself invariant to such a variation, the latent representation in this direction will vanish; (2) $\ell(f(h) + \epsilon\hat{n}^T\nabla f(h)) = \ell(f(h))$, which means that any variation that does not change the loss function value will vanish. This fact also means that the model will learn a compact representation: any irrelevant latent space will have zero variation. This is consistent with the often observed matching between the rank of the representation and the inherent dimension of the problem (Papyan, 2018; Ziyin, 2024). A major phenomenon of well-trained CNNs is that the latent representations learn to become essentially invariant to task-irrelevant features (Zeiler & Fergus, 2014; Selvaraju et al., 2017), an observation that matches the high-level features of the human visual cortex (Booth & Rolls, 1998). The CRH thus suggests a reason for these observed invariances.

This result also means that the invariances of the models achieved through the CRH are robust: the objective function is invariant to small noises in the irrelevant activations. While artificial neural networks rarely contain such noises in the activations, it is plausibly the case that biological neurons do suffer from environmental noises, and our result suggests a mechanism to achieve robustness against irrelevant noises or perturbation.

## 7.3 Reduction to Neural Collapse in Perfect Classification

We now prove that NC is equivalent to the CRH in an interpolating classifier. We focus on the first two properties here because NC3-4 are essentially consequences of NC1-2 (Rangamani & Banburski-Fahey, 2022). NC1 states that the inner-class variations of the penultimate representation vanishes. NC2 states that the average representation $\mu_c$ of the class $c$ is orthogonal to each other: $\mu_c^\top \mu_{c'} = \delta_{cc'}$. The ground truth model must be invariant to inner class variations in a classification task. Let $c$ denote the index of a class, it should be the case that the ground truth model $f'$ satisfies: $f'_c(x_c) = \zeta\mathbf{1}_c$, where $\mathbf{1}_c$ is a one hot vector on the $c$-th dimension and $\zeta > 0$ is an arbitrary scalar. Proposition 1 thus suggests that any model that recovers the ground truth must have such invariance in the output and in the latent representation $h$, which is exactly NC1.

Now, we show that when the CRH holds, a perfectly trained model must have neural collapse. Let $x_c$ denote any data point belonging to the $c$-class among a set of $C$ labels.

**Theorem 3.** *Consider a classification task and the penultimate layer postactivation $h_a$. If the model is quasi-interpolating: $f(x_c) = h_b = W h_a(x_c) = \zeta \mathbf{1}_c$, and the loss covariance is proportional to identity: $\mathbb{E}[\nabla_f \ell \nabla_f^\top \ell] \propto I$, then $h$ satisfies all four properties of neural collapse (NC1-NC4) if and only if $h$ satisfies the CRH.*

The assumption $\mathbb{E}[\nabla_f \ell \nabla_f^\top \ell] \propto I$ is empirically found to hold very well in standard classification tasks at the end of training. See Section C.3 for an experiment with Resnet on CIFAR-10, where the phenomenon of neural collapse is primarily observed.

## 7.4 Equivariance of CRH and Neural Feature Ansatz

A closely related hypothesis is the NFA, which is related to the feature learning in fully connected layers (Radhakrishnan et al., 2023; Beaglehole et al., 2023). Using our notation, the NFA states $W^\top W \propto \mathbb{E}[\nabla_{h_a} f \nabla_{h_a} f^\top]$. To see its connection to the CRH, note that it can be written as

$$W^\top W \propto W^\top \mathbb{E}[g_b g_b^\top] W = \mathbb{E}[g_a g_a^\top] = \mathbb{E}[(\nabla_{h_b} f) B (\nabla_{h_b} f^\top)], \tag{13}$$

where $B = \nabla_f \ell (\nabla_f \ell)^\top$. Therefore, the NFA is identical to the GWA (Eq. (5)) when $B \propto I$.

This is an instance of a consistency problem of the NFA: the NFA is not invariant to trivial redefinitions of the loss function and model. Let $f(x)$ be a model trained on loss function $\ell(f)$. Let us assume that NFA applies to this model. Now, let us construct a trivially redefined model and loss: $f'(x) := Z f(x)$ where $Z$ is an invertible matrix, and $\ell'(f) := \ell(Z^{-1} f)$. Now, for NFA to hold for $f$, we must have $W^\top W \propto \mathbb{E}[\nabla f \nabla^\top f]$, but for NFA to hold for $f'$, we need $W^\top W \propto \mathbb{E}[\nabla f Z Z^\top \nabla^\top f]$. These cannot hold simultaneously.

In contrast, the GWA is invariant to such redefinitions:

$$\nabla \ell(f) \nabla \ell(f) = \nabla \ell'(f') \nabla \ell'(f'). \tag{14}$$

Therefore, the CRH is more likely to be a fundamental law of learning as it is invariant to a subjective choice of basis. For this reason, one may also refer to the GWA as an "equivariant NFA" (eNFA). Furthermore, if we treat NFA as a special case of the forward alignment, the backward alignment relations imply a novel variant of the NFA, which can be referred to as the *forward NFA*: $W W^\top \propto \mathbb{E}[\nabla_{h_b} g \nabla_{h_b} g^\top]$. In this picture, the original NFA should thus be called the *backward NFA*. See Figure 8, which validates both forward and backward eNFA.

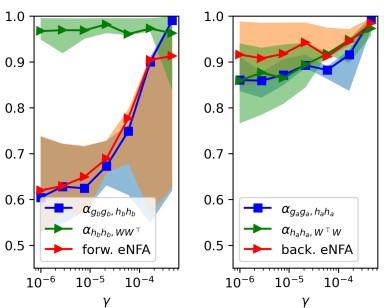

Figure 8: After the training of a six-layer fully connected network, all six relations (Eq. (3)-(5)) hold strongly at a large weight decay (**fc2**). For a small weight decay, at least one forward and backward relation holds strongly. The shaded region shows the variation across five hidden layers, and the solid lines show the median of these alignments. At a small $\gamma$, the best alignment is between $H_b$ and $W^\top W$ for the forward relation, and $G_a$ and $W W^\top$ for the backward, in agreement with the theoretical prediction.

## 8 Conclusion

In this work, we propose the Canonical Representation Hypothesis (CRH), a new perspective for studying the formation of representations in neural networks. The CRH suggests that representations align with the weights and gradients after training. It is a generalization of the neural collapse phenomenon for any fully connected layer in a neural network. In this view, representations are formed based on the degree and modes of deviation from the CRH. This deviation leads to the Polynomial Alignment Hypothesis (PAH), which posits that when the CRH is broken, distinct phases emerge in which the representations, gradients, and weights become polynomial functions of each other. A key future direction is to understand the conditions that lead to each phase and how these phases affect the behavior and performance of models. The CRH may also have biological implications as it implies that neural networks tend to learn an orthogonalized representation, which has been observed in the biological brain recently (Sun et al., 2025). The CRH may also have algorithmic implications. If representations align with the gradients (as in RGA), it might be possible to manually inject noise into neuron gradients to engineer specific structures in the model's representations. However, the CRH and PAH have several limitations. They apply only to fully connected layers, and a future step is extending them to other types of layers. Additionally, we have primarily focused on characterizing the final stage of representation formation. A more comprehensive theory of representation dynamics could lead to better training algorithms.

ACKNOWLEDGEMENTS

ILC acknowledges support in part from the Institute for Artificial Intelligence and Fundamental Interactions (IAIFI) through NSF Grant No. PHY-2019786. This work was also supported by the Center for Brains, Minds and Machines (CBMM), funded by NSF STC award CCF - 1231216.

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

## A MORE RELATED WORKS

The neural representation is a central object of study in both AI and neuroscience (Esser et al., 2020; Wang et al., 2018). The widespread use of techniques like t-SNE (Van der Maaten & Hinton, 2008) for analyzing the penultimate layer representation in deep learning highlights the importance of understanding representation space. However, the theory of how the representations are formed in neural networks is scarce. A recent work approached this problem from the angle of symmetries (Ziyin, 2024), showing that permutation symmetries in the latent layers lead to neuron merging and low rankness in the representation. Ziyin et al. (2024b) showed that after training, the latent representation is aligned with a linear transformation of the prediction residual. A recent work showed that the evolution of the representation during the initial stage of training have universal properties shared by different types of models (van Rossem & Saxe, 2024). Empirically, Roeder et al. (2021) and Huh et al. (2024) showed that the neural representations learned by different networks are essentially similar, which may offer a partial explanation for why the CRH seems to hold across different architectures. Also, an emergent field in neuroscience studies how the representations learned by neural networks closely resemble those of biological neurons (Rajalingham et al., 2018). These results suggest the existence of some fundamentally shared mechanisms of the formation of representations.

The finding that the neural networks find invariant structures is interesting and should be contrasted with the finding that sometimes the neural networks learns an invertible function (Zimmermann et al., 2021; Reizinger et al., 2024). This may have to do with the fact that both neural collapses have been found to emerge when there is nonnegligible strength of regularization (such as weight decay (Rangamani et al., 2021)). This suggests that some form of simplicity bias is required to make neural networks prefer invariant structures, which may be another interesting topic for future research.

## B THEORY AND PROOFS

### B.1 GRADIENT MOMENT IS DOMINATED BY GRADIENT COVARIANCE

We show that when there is a bias term in the layer, the expected neuron gradient must be negligible close to a local minimum:

$$h_b = W'h_a' = W h_a + \beta, \tag{15}$$

At the local minimum, we have

$$0 = \mathbb{E}[\Delta b] = -\eta(\mathbb{E}[\nabla_{h_b}\ell] + \gamma b) = \eta(\mathbb{E}[g_b] - \gamma b), \tag{16}$$

which implies that

$$\gamma b = \mathbb{E}[g_b]. \tag{17}$$

This implies that

$$\mathbb{E}[g_b]\mathbb{E}[g_b^\top] = O(\gamma^2), \tag{18}$$

which is negligible. This agrees with the experimental observation in Section C.2.

### B.2 PROOF OF THEOREM 2

We need a few lemmas. The following lemmas are quite general and apply to arbitrary symmetric matrices $A$ and $B$, which is clear from the lemma statement. Eventually, when we apply these lemmas, the specific $A$ and $B$ will be the $H$, $G$, $Z$ matrices we defined in the CRH statement.

**Lemma 1.** *Let $A$, $B$ be symmetric matrices such that $B = ABA$, then,*

$$\ker(A) \subseteq \ker(B) = I - B^0, \tag{19}$$

*and $B^0 A B^0$ is a projection matrix.*

*Proof.* Let $n$ be in the null space of $A$. We have

$$Bn = ABAn = 0. \tag{20}$$

Thus, $n$ is also in the null space of $B$. This means that the kernel of $A$ is in the kernel of $B$. Now, Let $P = B^0$ and $\tilde{D} := PDP$, we have that

$$B = \tilde{A}B\tilde{A}, \tag{21}$$

where the rank of $A$ is the same as the rank of $B$. This implies that there is an orthonormal matrix $O$ such that

$$B' = OBO^\top, \ A' = O\tilde{A}O^\top \tag{22}$$

are full rank, $A'$ is diagonal, and

$$B' = A'B'A'. \tag{23}$$

This implies that

$$B'_{ij} = a_i a_j B'_{ij}, \tag{24}$$

where $a_i$ is the $i$-th diagonal term of $A'$. Because $B'$ is full-rank. We have that for all $i$,

$$a_i = 1. \tag{25}$$

This implies that $\tilde{A} = O^\top A'O$ is an orthogonal projection matrix. Because it also has the same kernel as $B$, we have

$$\tilde{A} = B^0. \tag{26}$$

$\square$

The following two lemmas are straightforward to prove by multiplying $A^{-1}$ from the left and right.[4]

**Lemma 2.** *Let $A$, $B$ be symmetric matrices such that $A^2 = ABA$, then,*

$$A^0 = A^0 B A^0. \tag{27}$$

**Lemma 3.** *Let $A$, $B$ be symmetric matrices such that $A = ABA$, then,*

$$A^{-1} = A^0 B A^0. \tag{28}$$

**Lemma 4.** *If $\ker(A) = \ker(B)$, then $\ker(CAC^\top) = \ker(CBC^\top)$, for any symmetric $A$ and $B$ and arbitrary matrix $C$.*

*Proof.* For $n$ to be in the null space of $CAC^\top$, it must satisfy one of the following two conditions:

$$C^\top n = 0, \tag{29}$$

which implies that $n \in \ker(CBC^\top)$. Or

$$\sqrt{A}C^\top n = 0, \tag{30}$$

which implies that $C^\top n \in \ker(\sqrt{A}) = \ker(A) = \ker(B)$, which again implies that

$$n \in \ker(CBC^\top). \tag{31}$$

This finishes the proof. $\square$

Now, we are ready to prove Theorem 2.

*Proof.* (**Part 1**) Fix the direction to be forward or backward. Let $A, B, C$ be a permutation of three moment matrices. Then, that two alignments hold implies that

$$A \propto B, \tag{32}$$

and $B \propto C$. By transitivity, $A \propto B \propto C$. This proves part 1.

(**Part 2**) There are many combinations, which can be divided into a few types. We thus prove a prototype for each type of relation, and the rest can be derived in a similar but tedious manner. The key mechanism is that every forward relation implies a backward relation and vice versa. For example, if we know $\mathbb{E}[h_b h_b^\top] \propto \mathbb{E}[g_b g_b^\top]$, we also have

$$W\mathbb{E}[h_a h_a^\top]W^\top \propto \mathbb{E}[g_b g_b^\top], \tag{33}$$

---

[4]Recall that $A^{-1}$ is the pseudo inverse, and $A_0 = AA^{-1}$ is an orthogonal projector.

which implies that

$$W^\top W \mathbb{E}[h_a h_a^\top] W^\top W \propto W^\top \mathbb{E}[g_b g_b^\top] W = \mathbb{E}[g_a g_a^\top]. \tag{34}$$

Now, let $A$ be either $W^\top W$ or $WW^\top$, $B, C$ be some permutation of the gradient and neuron covariance.

Type 0: this is the simplest and most straightforward type of relations. First, consider relation 5 in the table. We have, by assumption,

$$H_a \propto Z_a, \tag{35}$$

$$WH_a W^\top = H_b \propto G_b. \tag{36}$$

Together, we have

$$WH_a W^\top \propto WZ_a W^\top = Z_b^2. \tag{37}$$

So,

$$Z_a^3 \propto W^\top G_b W = G_a. \tag{38}$$

This means that

$$H_a^3 \propto Z_a^3 \propto G_a. \tag{39}$$

Also, this implies that

$$H_b \propto Z_b^2 \propto G_b. \tag{40}$$

A similar proof derives relation 6 is thus not shown.

Type 1: $ABA \propto ACA \propto B \propto C$. Using the above lemmas and defining $\tilde{D} := PDP$, where $P = B^0 = C^0$, we obtain

$$\tilde{A} = B^0 = C^0. \tag{41}$$

Type 2: $ABA \propto A^2 \propto ACA$. This type of equation simply solves to

$$\tilde{B} \propto A^0 \propto \tilde{C}, \tag{42}$$

where the tilde is defined with respect to $A^0$.

Type 3: $B \propto A \propto ACA$. By the above lemmas, we have

$$A^{-1} \propto \tilde{C}. \tag{43}$$

We thus have

$$B \propto A \propto \tilde{C}^{-1}. \tag{44}$$

Type 4: $ABA \propto A^2 \propto C^2$. This implies that

$$\tilde{B} \propto A^0 \propto C^0, \tag{45}$$

where tilde is defined with respect to $P = A^0$.

Type 5: This type is a little strange, and so we explicitly solve them. There are two cases:

$$\mathbb{E}[g_a g_a^\top] \propto Z_a \propto Z_a^2 \tag{46}$$

$$\mathbb{E}[g_b g_b^\top] \propto Z_b \propto Z_b^2 \tag{47}$$

This directly implies that $Z_a$ and $Z_b$ are projection matrices. For $\mathbb{E}[hh^\top]$, we have that by definition

$$\mathbb{E}[h_b h_b^\top] \propto W \mathbb{E}[h_a h_a^\top] W^\top. \tag{48}$$

This means that the kernel of $\mathbb{E}[h_b h_b^\top]$ must contain the kernel of $WW^\top = Z_b$. Therefore, letting $P = \mathbb{E}[h_b h_b^\top]^0$

$$\mathbb{E}[h_b h_b^\top]^0 \propto PZ_b P \propto P\mathbb{E}[g_b g_b^\top]P. \tag{49}$$

Likewise, we have that

$$W\mathbb{E}[h_a h_a^\top]W^\top = \mathbb{E}[h_b h_b^\top], \tag{50}$$

and so

$$\ker(W\mathbb{E}[h_a h_a^\top]W^\top) = \ker(\mathbb{E}[h_b h_b^\top]) = \ker Z_b. \tag{51}$$

This implies that

$$\ker(Z_a \mathbb{E}[h_a h_a^\top] Z_a) = \ker Z_a. \tag{52}$$

This means that we have proved

$$\mathbb{E}[\tilde{h_a} h_a^\top]^0 = Z_a^0 = \mathbb{E}[g_a g_a^\top]. \tag{53}$$

A similar derivation applies to the case when

$$\mathbb{E}[h_a h_a^\top] \propto Z_a \propto Z_a^2 \tag{54}$$

$$\mathbb{E}[h_b h_b^\top] \propto Z_b \propto Z_b^2. \tag{55}$$

(**Part 3**) By the pigeonhole principle, two of either the forward or backward relations must be satisfied. This means that by part 2 of the theorem, there are two cases: (a) three forward relations are satisfied, (b) three backward relations are satisfied. Since the argument is symmetric in forward and backward directions, we focus on case (a).

Case (a). We have

$$\mathbb{E}[h_a h_a^\top] \propto Z_a \propto \mathbb{E}[g_a g_a^\top] = W^\top \mathbb{E}[g_b g_b^\top] W. \tag{56}$$

There are now three subcases depending on which of the backward relations are satisfied.

Case (a1): $Z_b \propto \mathbb{E}[g_b g_b^\top]$, which implies that

$$Z_a \propto \mathbb{E}[g_a g_a^\top] = W^\top \mathbb{E}[g_b g_b^\top] W \propto W^\top Z_b W = Z_a^2, \tag{57}$$

and so all the three forward matrices are (scalar multiples of) projections.

But $W^\top W$ and $W W^\top$ share eigenvalues and so $Z_b \propto \mathbb{E}[g_b g_b^\top]$ are also projection matrices. Now,

$$\mathbb{E}[h_b h_b^\top] = W \mathbb{E}[h_a h_a^\top] W^\top \propto W Z_a W^\top = Z_b^2, \tag{58}$$

which is also a projection matrix. This, in turn, implies that all three backward relations hold.

Case (a2): $\mathbb{E}[h_b h_b^\top] \propto \mathbb{E}[g_b g_b^\top]$. This implies that

$$Z_b^2 = W Z_a W^\top \propto W \mathbb{E}[h_a h_a^\top] W^\top = \mathbb{E}[h_b h_b^\top] \propto \mathbb{E}[g_b g_b^\top]. \tag{59}$$

In turn, this implies that

$$Z_a \propto \mathbb{E}[g_a g_a^\top] = W^\top \mathbb{E}[g_b g_b^\top] W \propto W^\top Z_b^2 W = Z_a^3. \tag{60}$$

This implies that $Z_a$ only contains zero and one as eigenvalues, which implies that $Z_a = Z_a^2 = Z_a^3$ is a projection. Similarly, $Z_b$ is a projection. Together, this implies that all six relations hold.

Case (a3): $Z_b \propto \mathbb{E}[h_b h_b^\top]$. This case is subtly different. Here,

$$\mathbb{E}[h_b h_b^\top] = W \mathbb{E}[h_a h_a^\top] W^\top = W Z_a W^\top = Z_b^2 = Z_b, \tag{61}$$

and so all three forward matrices are projections.

For the backward relations,

$$Z_b \propto \mathbb{E}[h_b h_b^\top] = W \mathbb{E}[h_a h_a^\top] W^\top \propto Z_b^2, \tag{62}$$

which is also a projection matrix. Now, we have essentially no relation for $\mathbb{E}[g_b g_b^\top]$ as it cannot be directly derived from any other quantity. But noting that

$$W^\top \mathbb{E}[g_b g_b^\top]^\top = \mathbb{E}[g_a g_a^\top] \propto Z_a, \tag{63}$$

one obtains that

$$P \mathbb{E}[g_b g_b^\top] P = P, \tag{64}$$

where $P = Z_b$. Therefore, when restricted to the subspace of $W$, $\mathbb{E}[g_b g_b^\top]$ is also a projection matrix.

For case (b), it is the same, except for the case when $Z_a \propto \mathbb{E}[g_a g_a^\top]$. When this is the case, the alignment happens with

$$P \mathbb{E}[h_a h_a^\top] P = P, \tag{65}$$

for $P = Z_a^0$.

**At local minimum**. At any local minimum or first-order stationary point, Lemma 5 applies, which implies that

$$\ker(Z_c) \subseteq \ker(\mathbb{E}[h_c h_c^\top]), \tag{66}$$

$$\ker(Z_c) \subseteq \ker(\mathbb{E}[g_c g_c^\top]), \tag{67}$$

for $c \in \{a, b\}$. This is because

$$Z_c \propto \mathbb{E}[g_c h_c^\top], \tag{68}$$

and for any $n \in \ker(\mathbb{E}[h_c h_c^\top])$, it holds with probability one that

$$h_c^\top n = 0, \tag{69}$$

and so

$$Z_c n = \mathbb{E}[h_c g_c^\top] n = 0. \tag{70}$$

The same argument applies to $g_c$.

Now, it suffices to prove the two subtle cases. For case (a3), we have that

$$P\mathbb{E}[g_b g_b^\top] P = P. \tag{71}$$

However, the local minimum condition implies that

$$\ker\mathbb{E}[g_b g_b^\top] \subseteq \ker(P). \tag{72}$$

This can only happen if $P\mathbb{E}[g_b g_b^\top] P = \mathbb{E}[g_b g_b^\top] = P$. Therefore, all six relations hold. The same applies to the other subtle case.

(**Part 4**) We have

$$\mathbb{E}[h_a h_a^\top] \propto Z_a \propto \mathbb{E}[g_a g_a^\top] = W^\top \mathbb{E}[g_b g_b^\top] W, \tag{73}$$

$$\mathbb{E}[h_b h_b^\top] = W \mathbb{E}[h_a h_a^\top] W \propto Z_b \propto \mathbb{E}[g_b g_b^\top]. \tag{74}$$

Plugging the forward relation into the backward relation, we obtain that

$$W Z_a W^\top \propto Z_b^2 \propto Z_b, \tag{75}$$

which implies that $Z_b$ is proportional to a projection matrix. This implies that $\mathbb{E}[h_b h_b^\top]$ and $\mathbb{E}[g_b g_b^\top]$ are also projection matrices.

Likewise, one can plug the backward relation into the forward, which implies that

$$Z_a^2 \propto Z_a, \tag{76}$$

and is thus proportional to a projection matrix. This completes the proof. $\qquad\square$

### B.3 Proof of Theorem 1

The proof will be divided into several theorems, each of which proves a claimed relation in Section 4. The following theorem proves Eq. 6.

**Theorem 4.** *Assume assumption 1. If $\mathbb{E}[\Delta H_a] = 0$ and $\mathbb{E}[\Delta H_b] = 0$, then,*

$$0 = z_b \mathbb{E}[g_b h_b^\top] + z_b \mathbb{E}[h_b g_b^\top] + \eta z_b^2 \mathbb{E}[g_b g_b^\top] - 2\gamma \mathbb{E}[\Delta(h_b h_b^\top)] + O(\eta^2 \gamma), \tag{77}$$

*where $z_b = \mathbb{E}\|h_a\|^2$.*

*Proof.* Let $B_a = h_a h_a^\top$, and $B_b = h_b h_b^\top$. Let $H_a = \mathbb{E}[B_b]$ and $H_b = \mathbb{E}[B_a]$. We consider an online learning setting. Let $x$ denote the input at time step $t$. Consider the time evolution of the feature covariance during the SGD training:

$$\Delta(h_b(x) h_b^\top(x)) = \Delta(W h_a h_a^\top W^\top) \tag{78}$$

$$= \Delta W B_a W^\top + W B_a \Delta W^\top + \Delta W B_a \Delta W^\top + O(\Delta B_a) \tag{79}$$

$$= -\eta(\nabla_{h_b}\ell h_a^\top + \gamma W) B_a W^\top - \eta W B_a (h_a \nabla_{h_b}^\top \ell + \gamma W^\top) + \eta^2(\nabla_{h_b}\ell h_a^\top + \gamma W) B_a (h_a \nabla_{h_b}^\top \ell + \gamma W^\top) \tag{80}$$

$$= -\eta(-\|h_a\|^2 g_b h_b^\top + \gamma B_b) - \eta(-\|h_a\|^2 h_b g_b^\top + \gamma B_b) + \eta^2 \|h_a\|^4 g_b g_b^\top + O(\eta^2 \gamma) \tag{81}$$

$$= \eta(\|h_a\|^2 g_b h_b^\top + \|h_a\|^2 h_b g_b^\top - 2\gamma B_b) + \eta^2 \|h_a\|^4 g_b g_b^\top. \tag{82}$$

Taking expectation of both sides, we obtain that

$$0 = z_b \mathbb{E}[g_b h_b^\top] + z_b \mathbb{E}[h_b g_b^\top] + \eta z^2 \mathbb{E}[g_b g_b^\top] - 2\gamma H_b, \tag{83}$$

where $z_b = \mathbb{E}\|h_a\|^2$. $\qquad\square$

Similarly, one can show that the pre-activations follow a similar relationship.

**Theorem 5.** *Assume assumption 1. If $\mathbb{E}[\Delta G_a] = 0$ and $\mathbb{E}[\Delta G_b] = 0$, then,*

$$z_a(\mathbb{E}[h_a g_a^\top] + \mathbb{E}[g_a h_a^\top]) + \eta z_a^2 \mathbb{E}[h_a h_a^\top] = 2\gamma \mathbb{E}[g_a g_a^\top], \tag{84}$$

*where we have defined $z_a = \mathbb{E}[\|g_b\|^2]$*

*Proof.* We have

$$\Delta(g_a g_a^\top) = \Delta(W^\top g_b g_b^\top W) \tag{85}$$

$$= \Delta W^\top g_b g_b^\top W + W^\top g_b g_b^\top \Delta W + \Delta W^\top g_b g_b^\top \Delta W + O(\|\Delta g_b g_b^\top\|) \tag{86}$$

$$= \eta(z_a h_a g_a^\top - \gamma g_a g_a^\top) + \eta(z_a h_a g_a^\top - \gamma g_a g_a^\top)^\top + \eta^2 z_a^2 h_a h_a^\top + O(\eta^2 \gamma), \tag{87}$$

where we have used the relations $g_a = W^\top g_b$ and $\Delta W = \eta(g_b h_a^\top - \gamma W)$. We have also defined $z_a = \|g_b\|^2 = \mathbb{E}[\|g_b\|^2]$.

Taking expectation, this leads to

$$z_a(\mathbb{E}[h_a g_a^\top] + \mathbb{E}[g_a h_a^\top]) + \eta z_a^2 \mathbb{E}[h_a h_a^\top] = 2\gamma \mathbb{E}[g_a g_a^\top]. \tag{88}$$

$\square$

The following lemma proves the balance condition at a local minimum.

**Lemma 5.** *(First-order stationary point condition / local minimum balance.) At any stationary point of the loss function, we have*

$$\mathbb{E}[g_b h_b^\top] = \mathbb{E}[g_b h_b^\top]^\top = \gamma W W^\top, \tag{89}$$

$$\mathbb{E}[g_a h_a^\top] = \mathbb{E}[g_a h_a^\top]^\top = \gamma W^\top W. \tag{90}$$

*Proof.* Close to a stationary point, we have

$$0 = \mathbb{E}[\Delta W] = \eta(\mathbb{E}[g_b h_a^\top] - \gamma W). \tag{91}$$

Multiplying $W^\top$ from the left and the right, we obtain, respectively:

$$0 = \mathbb{E}[g_b h_a^\top]W^\top - \gamma W W^\top = \mathbb{E}[g_b h_b^\top] - \gamma W W^\top, \tag{92}$$

$$0 = W^\top \mathbb{E}[g_b h_a^\top] - \gamma W^\top W = \mathbb{E}[g_a h_a^\top] - \gamma W^\top W, \tag{93}$$

where we have used the definition $h_b = W h_a$ and the chain rule $g_a = W^\top g_b$.

This condition implies that

$$\mathbb{E}[g_b h_b^\top] = \mathbb{E}[g_b h_b^\top]^\top = \gamma W W^\top, \tag{94}$$

$$\mathbb{E}[g_a h_a^\top] = \mathbb{E}[g_a h_a^\top]^\top = \gamma W^\top W. \tag{95}$$

$\square$

Now, we derive the stationarity condition for $\Delta W W^\top$ and $\Delta W^\top W$.

**Lemma 6.** *(Stationary alignment of parameter-outer product.) If at a local minimum and assuming Assumption 1, then,*

1. *if $\mathbb{E}[\Delta(W W^\top)] = 0$,*

$$z_b \mathbb{E}[g_b g_b^\top] = \gamma^2 W W^\top. \tag{96}$$

2. *if $\mathbb{E}[\Delta(W^\top W)] = 0$,*

$$z_a \mathbb{E}[h_a h_a^\top] = \gamma^2 W^\top W. \tag{97}$$

*Proof.* We have

$$0 = \mathbb{E}[\Delta(WW^\top)] = \mathbb{E}[\Delta W]W^\top + W\mathbb{E}[\Delta W^\top] + \mathbb{E}[\Delta W \Delta W^\top] \tag{98}$$

$$= 0 + \mathbb{E}[\Delta W \Delta W^\top] \tag{99}$$

$$= \mathbb{E}[(g_b h_a^\top - \gamma W)(g_b h_a^\top - \gamma W)^\top] \tag{100}$$

$$= z_b \mathbb{E}[g_b g_b^\top] - \gamma \mathbb{E}[g_b h_b^\top] - \gamma \mathbb{E}[g_b h_b^\top]^\top + \gamma^2 WW^\top \tag{101}$$

$$= z_b \mathbb{E}[g_b g_b^\top] - \gamma^2 WW^\top \tag{102}$$

where we have used Eq. (89) in the last line.

Likewise, when $\mathbb{E}[\Delta W^\top W] = 0$,

$$0 = \mathbb{E}[\Delta(W^\top W)] = \mathbb{E}[(g_b h_a^\top - \gamma W)^\top (g_b h_a^\top - \gamma W)] \tag{103}$$

$$= z_a \mathbb{E}[h_a h_a^\top] - \gamma \mathbb{E}[h_a g_a^\top] - \gamma \mathbb{E}[h_a g_a^\top]^\top + \gamma^2 W^\top W \tag{104}$$

$$= z_a \mathbb{E}[h_a h_a^\top] - \gamma^2 W^\top W, \tag{105}$$

where we have defined $z_a = \mathbb{E}[\|g_b\|^2]$.

$\square$

Now, we are ready to prove Theorem 1.

*Proof.* The above two lemmas imply that

$$\frac{2z_b}{\gamma} \mathbb{E}[g_b g_b^\top] = \mathbb{E}[g_b h_b^\top] + \mathbb{E}[g_b h_b^\top]^\top. \tag{106}$$

This relation can be substituted into Eq. (6) to obtain

$$z_b^2(\eta\gamma + 2)\mathbb{E}[g_b g_b^\top] = 2\gamma^2 H_b = z_b(\eta\gamma + 2)\gamma^2 WW^\top. \tag{107}$$

Likewise,

$$\frac{2z_a}{\gamma} \mathbb{E}[h_a h_a^\top] = \mathbb{E}[h_a g_a^\top] + \mathbb{E}[h_a g_a^\top]^\top. \tag{108}$$

This relation can be plugged into Eq. (84) to obtain that

$$z_a^2(\eta\gamma + 2)\mathbb{E}[h_a h_a^\top] = 2\gamma^2 \mathbb{E}[g_a g_a^\top] = z_a(\eta\gamma + 2)\gamma^2 W^\top W. \tag{109}$$

This completes the proof. $\square$

### B.4 PROOF OF PROPOSITION 1

*Proof.* We first prove that part (1) implies part (2). By assumption, we have that $\mathbb{E}[hh^\top] \propto \mathbb{E}[gg^\top]$, and so[5]

$$E[hh^\top]\hat{n} \propto \mathbb{E}[gg^\top]\hat{n} \tag{110}$$

$$= \mathbb{E}[\nabla_h \ell \nabla_h^\top \ell]\hat{n} \tag{111}$$

$$= 0, \tag{112}$$

where the last line follows from the fact that

$$\ell(f(h + \epsilon\hat{n})) - \ell(f(h)) = \epsilon\hat{n}^\top \nabla_h \ell(h) + O(\epsilon^2) = O(\epsilon^2), \tag{113}$$

which is possible only if $\hat{n}^\top \nabla_h \ell(h) = 0$. One can derive the other two relations simply using the definition of CRH: $\mathbb{E}[hh^\top] \propto \mathbb{E}[gg^\top] \propto Z$.

For the backward direction, we have that

$$E[hh^\top]\hat{n} = \mathbb{E}[gg^\top]\hat{n} = 0. \tag{114}$$

---

[5]Note that the proof still works if we replace the second moments with the covariances.

This implies that $n^\top g = 0$ with probability 1. Now,

$$\ell(f(h+\epsilon\hat{n})) - \ell(f(h)) = \epsilon\hat{n}^\top g + O(\epsilon^2) \tag{115}$$

$$= O(\epsilon^2). \tag{116}$$

The proof is complete.

$\square$

### B.5 Proof of Theorem 3

As NC4 is a trivial consequence of NC1-3, we focus on NC1-3 here. For notational simplicity, we state neural collapse in the case when there is no bias in the last layer. The first three properties are defined as, at the end of training

1. NC1: $h(x_c) = \mu_c$, where $x_c$ is any data point in class $c$;
2. NC2: $\mu_c^\top \mu_{c'} = \delta_{cc'}$;
3. NC3: $W^\top W = \sum_c^C \mu_c \mu_c^\top$.

*Proof.* We first prove that NC1-4 implies the CRH. When NC1 holds,

$$h_a(x_c) = \mu_c. \tag{117}$$

This means that

$$H_a \propto \sum_c \mu_c \mu_c^\top. \tag{118}$$

By NC2, we have

$$\mu_c^\top \mu_{c'} = \delta_{cc'}. \tag{119}$$

This means that $H_a$ is proportional to an orthogonal projection.

By NC3, we have that

$$W^\top W \propto \sum_c \mu_c \mu_c^\top. \tag{120}$$

Additionally,

$$G_a = W^\top G_b W = W^\top W, \tag{121}$$

where we have used the assumption that $G_b = \mathbb{E}[\nabla_f \ell \nabla_f^\top \ell] = I$.

Together, this implies the backward CRH

$$H_a \propto G_a \propto Z_a. \tag{122}$$

For the forward CRH, because $WW^\top \in \mathbb{R}^{C \times C}$ has rank $C$ and they have equal eigenvalues, it must be proportional to identity:

$$Z_b \propto I. \tag{123}$$

By the interpolation assumption, we also have

$$\mathbb{E}[h_b h_b^\top] \propto \sum_c^C \mathbf{1}_c \mathbf{1}_c^\top \propto I. \tag{124}$$

Therefore, we have proved the forward CRH. This proves one direction of the theorem.

Now, we prove that the CRH implies NC1-NC3. We first prove NC1. By the (backward) alignment hypothesis for the last layer, we have

$$H_a \propto G_a = W^\top G_b W = W^\top W, \tag{125}$$

where $W$ is the weight matrix for the last layer. This means that there exists an orthonormal matrix $U$ such that

$$W \propto U\sqrt{H_a}, \tag{126}$$

and that the rank of $H_a$ must be $C$. By the interpolation hypothesis, it must be the case that

$$W h_c = \mathbf{1}_c, \tag{127}$$

which implies that for a fixed $c$. This implies that

$$h_c = z_c + v, \tag{128}$$

where $z_c$ is a constant vector and $Wv = 0$. However, by Proposition 1, $v$ must vanish, and so $h_c = z_c = \mu_c$. This proves NC1.

This means that

$$Z_a \propto H_a \tag{129}$$

is essentially a projection matrix. So,

$$W h_a(x_c) = U H_a h_a(x_c) = U h_a(x_c) = \mathbf{1}_c. \tag{130}$$

This implies that in turn, $h_a(x_c) = \mu_c = U_{c:}$. This proves NC3. Due to the orthogonality of $U$, we have that

$$\mu_c^\top \mu_{c'} = U_{c:}^\top U_{c':} = \delta_{cc'}. \tag{131}$$

This proves NC2. The proof is complete. $\qquad\square$

## C  EXPERIMENTS

### C.1  EXPERIMENTAL DETAILS

**fc1:**  Fully connected neural networks trained on a synthetic dataset that we generated using a two-layer teacher network. This experiment is used for a controlled study of the effect of different hyperparameters. To control the variables, we consider a synthetic task where the input $x \in \mathbb{R}^{100}$ is sampled from an isotropic Gaussian distribution, and the label generated by a nonlinear function of the form: $y(x) = \sum_{i=1}^{100} u_i^* \sin((W_i^*)^\top x + b_i^*) \in \mathbb{R}$, where $u^*$, $w_i^*$, and $b_i^*$ are fixed variables drawn from a Gaussian distribution. In this form, the target function can be seen as a two-layer network with sin activation. Our model is a fully connected neural network trained with SGD in an online fashion, and the representations are computed with unseen data points. Unless specified to be the independent variable, the controlled variables of the experiment are: depth of the network ($D = 4$), the width of the network ($d = 100$), weight decay strength ($\gamma = 2 \times 10^{-5}$), minibatch size ($B = 100$).

**fc2:**  Here, we choose a high-dimensional setting where the teacher net is given by $y(x) = \sum_{i=1}^{100} u_i^* \sin((W_i^*)^\top x + b_i^*) \in \mathbb{R}$, where $u_i^* \in \mathbb{R}^{100}$, $w_i^* \in \mathbb{R}^{100}$, and $b_i^* \in \mathbb{R}$ are fixed variables drawn from a Gaussian distribution. In this form, the target function can be seen as a two-layer network with sin activation. The distribution of $x$ is controlled by an independent variable $\phi_x$ such that $x' \sim \mathcal{N}(0, I_{100})$, and $x = (1 - \phi_x)Z + \phi_x I_{100}$, where $Z$ is a zero-one matrix generated by a Bernoulli distribution with probability $0.8$. When $\phi_x$ is small, the input features are thus highly correlated to each other, and the covariance matrix deviates far from $I$. The training proceeds with SGD with a learning rate of $0.1$ with momentum $0.9$ and $\gamma = 10^{-4}$ for $10^5$ steps when the loss function value has stopped decreasing. The training proceeds with a batch size of $100$. All the expectation and covariance matrices are estimated using $3000$ independently sampled data points. The trained model is a 5-hidden layer fully connected network with the ReLU activation.

**res1:**  ResNet-18 (11M parameters) for CIFAR-10; we apply the standard data augmentation techniques and train with SGD with a learning rate $0.01$, momentum $0.9$, cosine annealing for 200 epochs, and batch size 128. The model has four convolutional blocks followed by two fully connected layers with ReLU activations. The model has 11M parameters and achieves $94\%$ test accuracy after training, in agreement with the standard off-the-shelf ResNet-18 for the dataset.

**res2:**  ResNet-18 for self-supervised learning tasks with the CIFAR-10/100 datasets. The model is the same as res1, except that the last fc layer output becomes 128-dimensional, which is known as the projection dimension in SSL. The training follows the default procedure in the original paper (Chen et al., 2020), proceeding with a batch size of $512$ and $\gamma = 5 \times 10^{-5}$ for 1000 epochs.

**llm:**  a six-layer transformer (100M parameters) trained on the OpenWebText (OWT) dataset (Gokaslan & Cohen, 2019); the number of parameters of this model matches the smallest version of GPT2. The model has six layers with eight heads per layer, having 100M trainable parameters in total. For each experiment, we train with Adam with a weight decay strength of $1 \times 10^{-4}$ for $10^5$ iterations, when the training loss stops changes significantly. Since every representation has three dimensions: data $N$, token $T$, and feature $F$, we treat each token as if they are a separate sample in computing the covariances. Namely, we contract the representation tensor along the data and token dimension, resulting in a $F \times F$ covariance matrix.

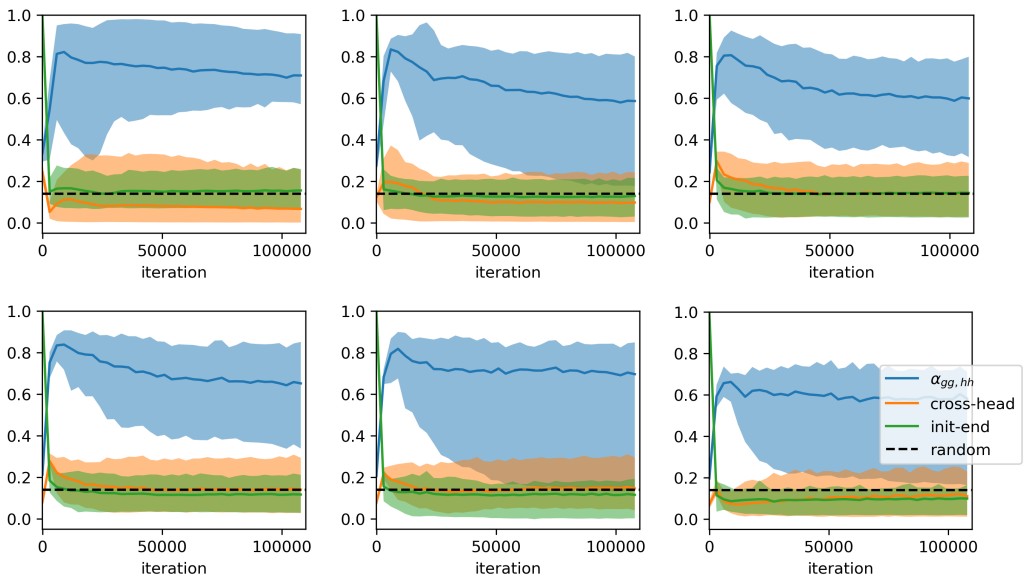

Figure 9: Alignment of the matrices $\mathbb{E}[hh^\top]$ and $\mathbb{E}[gg^\top]$. The experimental setting is the same as the LLM experiment.

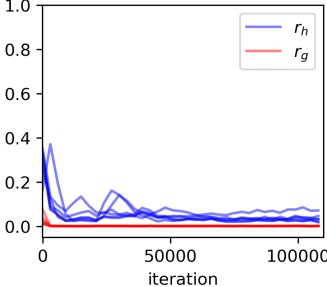

Figure 10: The ratios of the traces of the second moment matrices for transformer: $r_g = \mathrm{Tr}[\mathbb{E}[g]\mathbb{E}[g^\top]]/\mathrm{Tr}[\mathbb{E}[gg^\top]]$ and $r_g = \mathrm{Tr}[\mathbb{E}[h]\mathbb{E}[h^\top]]/\mathrm{Tr}[\mathbb{E}[hh^\top]]$. We see that $r_g$ essentially converges to zero, which means that $\mathbb{E}[gg^\top] = \mathrm{cov}(g, g)$ at the end of training. $r_h$ is generally non-zero but is essentially negligible. The experiment setting is the same as the LLM experiments.

## C.2 SECOND MOMENT ALIGNMENTS

This section shows the results for the alignment of the matrices $\mathbb{E}[hh^\top]$ and $\mathbb{E}[gg^\top]$. See Figure 9. The results are qualitatively similar to the result for the alignment between $\mathrm{cov}(h, h)$ and $\mathrm{cov}(g, g)$, but with a larger variation. The reason for the similarity is that it is often the case that the covariance term dominates the second moments at the end of training. See Figure 10.

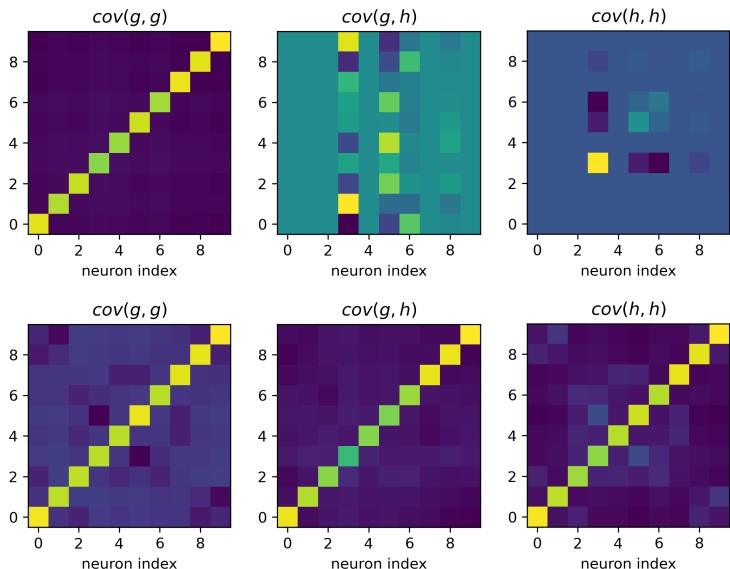

Figure 11: Representation of the output layer of Resnet18. Essentially, these are the covariances of the output at the end of training. **First Row:** Initialization. **Second Row**: End of training.

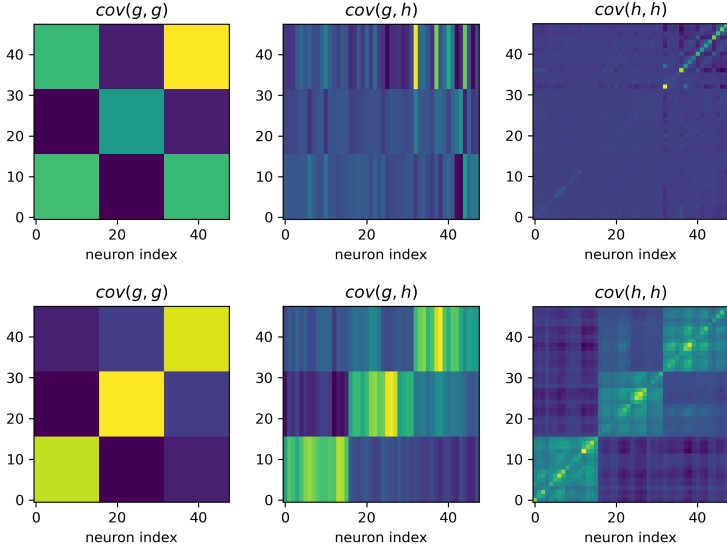

Figure 12: Representation covariance of the last convolution block at initialization (**upper**) and end of training (**lower**).

## C.3    REPRESENTATIONS OF RESNET18

See Figure 12 for the representations of the last convolutional layer of Resnet18 before and after training on CIFAR-10. See Figure 11 for the representations of the output layer. Interestingly, for the classification task, both $\mathrm{cov}(g, g)$ and $\mathrm{cov}(h, h)$ become proportional to the identity.

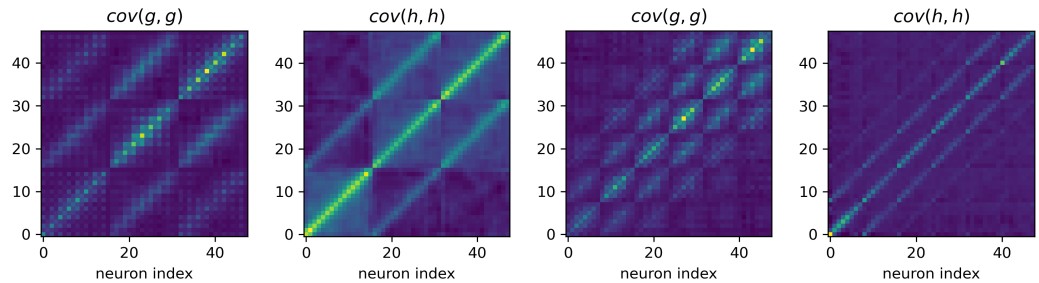

Figure 13: Examples of representation for Resnet-18 after a self-supervised contrastive training. **Left**: second convolution block representation, **Right**: penultimate convolution block representation.

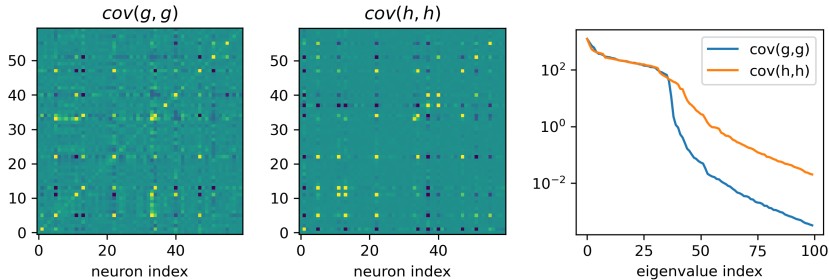

Figure 14: Examples of the representation learned by the transformer in the third layer. **Left-Mid**: examples. **Right**: The spectra of the two matrices are exactly the same for the leading eigenvalues. The difference is mainly in the smaller eigenvalues, and this difference gets smaller as the training proceeds.

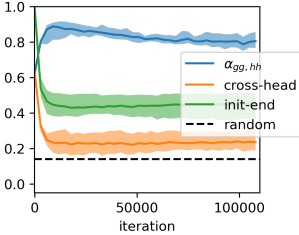

Figure 15: The alignment of feature and gradient covariance ($\alpha_{gg,hh}$) remains high during most of the training (**llm**). The shaded region shows the variation across 8 different heads in the same layer.

### C.4 REPRESENTATIONS OF SELF-SUPERVISED LEARNING

See Figure 13 for the representations of the last and the penultimate convolutional layers. They have significant alignments, but the agreement is perfect. For fully connected layers, the alignment is much better (see the main text; examples not shown).

### C.5 LARGE LANGUAGE MODEL

See Figure 14 and 15.

## C.6 FULLY CONNECTED NETS

See Figure 16 and 17. We see that the alignment effect is significant for both SGD and Adam. Also, see Figure 18 for the effect of having different depths.

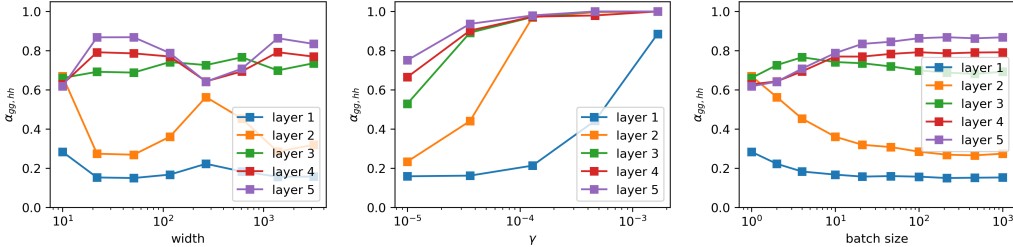

Figure 16: The effect of the width, weight decay $\gamma$, and batch size on the alignment for a fully connected network. The training proceeds for $10^5$ iterations, when the training stops decreasing significantly. The training proceeds with **SGD**.

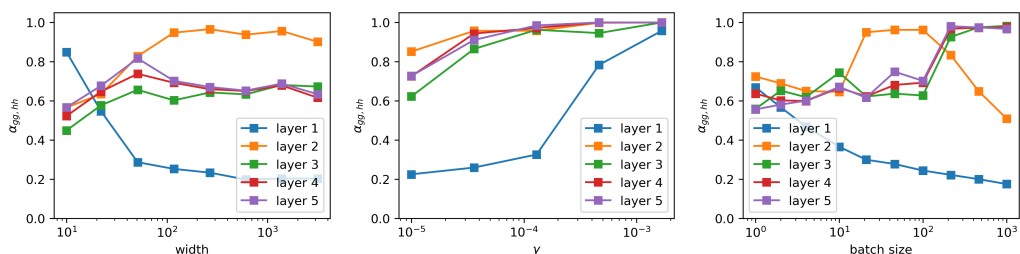

Figure 17: Same as the previous figure, except that the training proceeds with **Adam**.

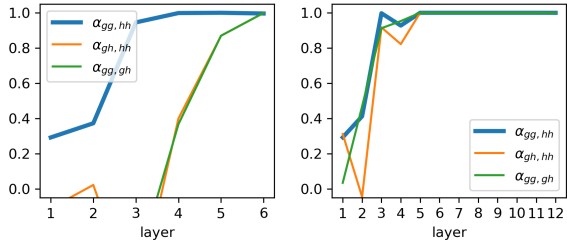

Figure 18: Alignment of of different layers of a fully connected ReLU network at different layers (layer 0 is the input layer). **Left**: a 6-layer network. **Right**: a 12-layer network.

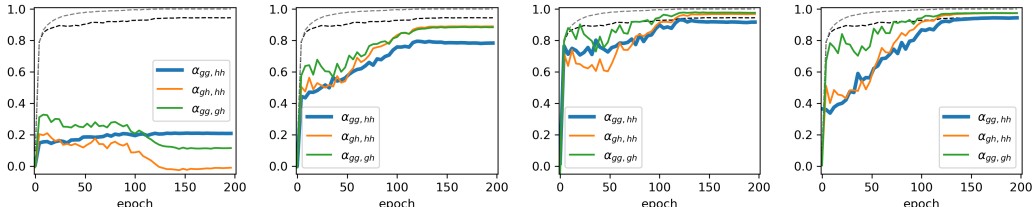

Figure 19: Evolution of $\alpha$ during training for different layers of Resnet-18 on CIFAR-10. For reference, the training accuracy (grey) and testing accuracy (black) are shown in the dashed line. **Left** to **Right**: (1) penultimate convolution block representation, (2) last convolution block representation, (3) penultimate fully connected layer, (4) output layer.

## C.7 CRH IN RESNET-18

See Figure 1.

## C.8 CRH AND PAH IN FULLY CONNECTED NETS

See Figure 20. The task is the same as other fully connected net experiments. The model is a 4-hidden-layer tanh net with the same width.

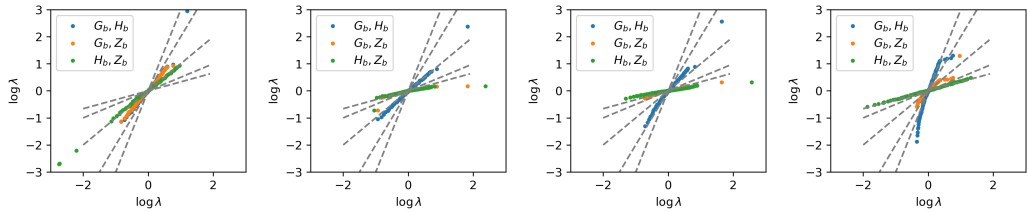

Figure 20: The alignment scalings in fully connected nets. The dashed lines show power laws with exponents $1/3,\ 1/2,\ 1,\ 2,\ 3$, respectively.

## C.9 STATIONARITY OF THE COVARIANCE MATRIX

See Figure 21 for the evolution of $\Delta\mathrm{cov}(h, h)$ to zero for three convolutional layers (0-2) and the two fully connected layers (3-4) in a Resnet-18 during training.

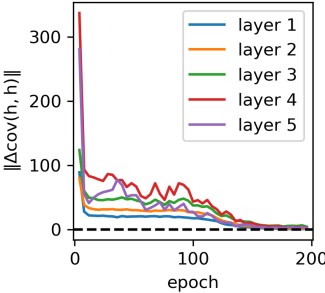

Figure 21: The change in the representation covariance converges to (near) zero at the end of training. The model is Resnet-18 trained on CIFAR-10 with standard SGD.

## C.10 FULL FIGURE TO FIGURE 4

See Figure 22.

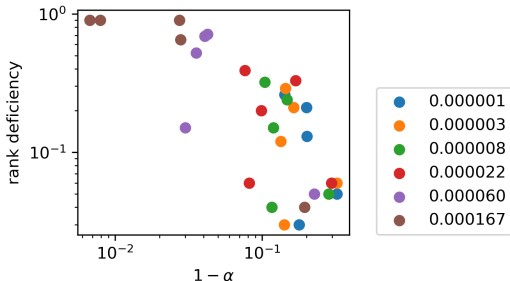

Figure 22: This is the same figure as Figure 4. The legend shows the weight decay value used for each experiment. Dots with the same color come from the five different hidden layers for the model trained at this specific weight decay value.

# D  MEANING OF THE PAH

## D.1  LINEAR REGRESSION AND PHASES OF THE PAH

In some sense, the phases of the PAH already appear implicitly in many standard models. Consider a linear regression problem:

$$W^* = \min_{W} \mathbb{E}\|Wx - y\|^2, \tag{132}$$

where $\mathbb{E}$ denotes averaging over the training set. Its solution is well known:

$$(W^*)^\top = \mathbb{E}[xx^\top]^{-1}\mathbb{E}[xy]. \tag{133}$$

From the perspective of the PAH, this solution can be seen as a composition of two functional layers: $W^* = V_1 V_2$, where the first layer is $V_1 = \mathbb{E}[xx^\top]^{-1/2}$ and the second layer is $V_2 = \mathbb{E}[\tilde{x}y]$, and $\tilde{x} = \mathbb{E}[xx^\top]^{-1/2}x$.

The first layer normalizes the input distribution and is apparently related to the PAH because one can identify

$$H_a = \mathbb{E}[xx^\top] \tag{134}$$

as the input representation to the layer, and so

$$\mathbb{E}[xx^\top]^{-1} = (W^*)^\top W^* = Z_a, \tag{135}$$

which implies that

$$H_a^{-1} = Z_a, \tag{136}$$

which can be identified as the 7th phase of the PAH in Table 1.

## D.2  POTENTIAL IMPLICATIONS OF THE PAH

Here, we discuss some possible and interesting meanings of the phases of the PAH. Validating these intuitions could be of great interest.

One way the phase could imply, as we suggested in the manuscript, is that a positive relation between $H_a$ and $Z_a$ could imply a low-rank mechanism, where larger eigenvalues of $H_a$ are expanded while smaller eigenvalues are suppressed. The fact that neural networks learn these low-rank representations is consistent with common observation. For example, see Kobayashi et al. (2024). Another implied mechanism is representation normalization. This happens when $H_a \propto Z_a^{-1}$, which means that $H_b$ will be normalized after this weight. See the discussion about linear regression in Section D.1 for how this phase exists implicitly, even in linear regression.

Another kind of interesting phase is where the alignment condition implies gradient vanishing or explosion problems, which have been well-known for decades (Kanai et al., 2017; Hochreiter, 1998). Noticing that $g_a = W^T g_b$, it is naturally the case that if $Z_b$ and $G_b$ are aligned with a positive exponent, the gradient in the previous layer will face an explosion and vanishing problem simultaneously (as larger eigenvalues become larger). In contrast, the phase $Z_b \propto G_b^{-1}$ seems to be the ideal training phase as it implies that the gradient in the previous will be normalized. Identifying the causes of these phases is also of great future interest.

