# OpenReview forum: "Formation of Representations in Neural Networks"
_ICLR.cc/2025/Conference — ICLR 2025 Spotlight_

### Official Review · Reviewer_FkCt · 2024-11-02

**Soundness:** 3
**Presentation:** 3
**Contribution:** 2
**Rating:** 6
**Confidence:** 2

**Summary:**

The paper proposes and studies the 'canonical representation hypothesis'. They study the evolution of weights, representations and gradients in neural networks and establish that these quantities become aligned. The paper includes theoretical derivations and experimental measurements that study how the CRH is fulfilled or broken.

**Strengths:**

The paper makes an effort to be thorough in examining all possible combinations of the forward and backward alignment relations. Moreover, they include a significant number of theoretical results which distinguishes this paper from most work in machine learning.

**Weaknesses:**

Why should we be interested in the alignment of weights, gradients and representations? How does this contribute to the bigger picture of understanding why deep neural network work. If the authors provided a clear motivation there would be much more incentive to dig through the notation and derivations.

**Questions:**

23: emergence + of

44: Furthermore, ... - this sentence makes no sense without more context

49: How does the CRH have anything to do with 'interpretable' solutions?

Related work, you might want to include: https://arxiv.org/abs/2007.00810

68: weight matrices evolve during training? This seems rather tautological, what is the point?

117: missing verb?

149: In practice, it is often not the case that stationarity is reached. Comment?

159-160: Can you explain?

217: Neural collapse / invariance to irrelevant features seems to be at odds with these works (https://proceedings.mlr.press/v139/zimmermann21a.html; https://arxiv.org/abs/2410.21869). Comment?

232: Why is this 'surprising' if it is exactly what the theory predicts?

---

> ### Author Response · Authors · 2024-11-20
> **Rebuttal Part 1**
>
> Thanks for your constructive feedback. We answer both the weaknesses and questions below.
>
> **Why should we be interested in the alignment of weights, gradients and representations? How does this contribute to the bigger picture of understanding why deep neural network work. If the authors provided a clear motivation there would be much more incentive to dig through the notation and derivations.**
>
> Thanks for giving us the chance to answer this important question, which we only concisely touched on in the manuscript due to space constraints. Please see the **Relevance and Implication** part of the summary rebuttal for our answer to this question. We believe that the study of these alignment relations can have broad and interdisciplinary implications for (1) the theory and science of deep learning, (2) the practice of deep learning, (3) physics, and (4) neuroscience.
>
> **23: emergence + of**
>
> Thanks for pointing out the typo. We have corrected this in the update.
>
> **44: Furthermore, ... - this sentence makes no sense without more context**
>
> This sentence discusses what happens when the CRH breaks. We have improved this sentence.
>
> **49: How does the CRH have anything to do with 'interpretable' solutions?**
>
> This is a good question. Some alignment relations are "interpretable" in the sense that they reveal what it means for a hidden neuron to have a high activation magnitude or variability. One primary example we gave in the manuscript is the RGA (representation-gradient alignment), whose diagonal terms imply that the neuron variations (across different inputs) are aligned with the neuron gradients, which are often "interpreted" as the importance/saliency of the neuron. Therefore, the CRH can indeed help us better interpret and understand the firing of the hidden neurons.
>
>
> **Related work, you might want to include: https://arxiv.org/abs/2007.00810**
>
> Thanks for this interesting reference. This work shows that representations learned by different models are similar to each other after an affine transformation. This may well offer an explanation for our observation that the CRH holds quite independent of architectural details. We have included a reference to it.
>
> **68: weight matrices evolve during training? This seems rather tautological, what is the point?**
>
> Thanks for pointing this out. This is a typo with a missing phrase. We intended to say that: "Another related phenomenon is the neural feature ansatz (NFA) (Radhakrishnan et al., 2023), which shows that the weight matrices of fully connected layers evolve according to the output gradient outer product during training. "
>
> **117: missing verb?**
>
> Yes. We missed the verb "hold". It should be stated as "That these relations simultaneously hold for any fully connected layer will be referred to as the canonical representation hypothesis (CRH)."
>
> **149: In practice, it is often not the case that stationarity is reached. Comment?**
>
> Thanks for raising this question. This is not easy to answer because the answer is much more subtle than one intuitively expects. The answer is not yes or no. To our knowledge, there is no thorough scientific study of this problem, so our answer is based on our experimental observations and educated guesses.
>
> Experimentally, these matrices do reach (approximate) stationarity, as demonstrated in Figure 21 in Section C9. We see that at the end of the training, the representation covariance reaches stationarity in all layers of ResNet18 trained on CIFAR10. An additional intriguing observation from this figure is that the convolutional layers reach stationarity as well -- this could imply that some forms of CRH (after taking the weight-sharing effect into account) might apply to the convolutional layers as well, and this could be an interesting future step.
>
> Conceptually, we believe that a better question to ask is, "To what extent the stationarity is reached?" In many dynamical systems, it is possible for some subspace to reach stationarity while the entire motion does not. For a common example, the earth-orbiting-the-sun process does not reach full stationarity, yet the angular momentum does not change in time and reaches stationarity. It is thus reasonable to expect that similar things happen in deep learning -- it is certainly not true that all parameters reach stationarity, but it is possible that some function or statistics of the parameters reach stationarity (at least in expectation). This seems to be the case, given the experimental results in Figure 21.
>
> **159-160: Can you explain?**
>
> Thanks for this question. The statements in 159-160 are proved in detail in Lemma 5 and Lemma 6. The mechanism is simply to check the zero gradient condition, which leads to $\mathbb{E}[\Delta W] =0$. We have added a reference to the proofs in the main text.

---

> ### Author Response · Authors · 2024-11-20
> **Rebuttal Part 2**
>
> **217: Neural collapse / invariance to irrelevant features seems to be at odds with these works (https://proceedings.mlr.press/v139/zimmermann21a.html; https://arxiv.org/abs/2410.21869). Comment?**
>
> Thanks for this question. You seem to suggest the following (are we correct?): neural collapse seems to suggest that the data processing in the network is not invertible, and hence, there seems to be a contradiction between these two works, which suggests that in certain scenarios, the data processing is invertible.
>
> However, we do not think there is really any contradiction. On the one hand, even if neural collapse happens, the data process might still look like they are "approximately" invertible. This agrees with the empirical findings in these works, where the inversion was almost never found to be perfect. On the other hand, the settings where one has neural collapse may be exclusive to the settings where the learning is invertible. For example, neural collapse is easier to happen when there is a regularization term (weight decay), and the same applies to CRH, whereas these two prior works do not seem to discuss the role of weight decay at all.
>
> In any case, we do not think there is any direct contradiction, although an interesting future direction would be to identify the precise conditions for the CRH to happen or not to happen. We have included these references and their discussion to the manuscript (Section A).
>
>
> **232: Why is this 'surprising' if it is exactly what the theory predicts?**
>
> Thanks for this subtle but interesting question. It is empirically surprising because the theory only says that if the CRH is (only) partially broken, then there would be power laws. The theorem does not say, however, when the CRH is partially broken, which remains an interesting open problem to be studied in the future. It could well be possible that this partial breaking is almost never observed -- for example, this would be the case if the CRH is almost always fully broken. That one observes these power laws "surprisingly" shows that a partial breaking of the CRH is common -- which the theory does not predict.

---

> > ### Comment · Reviewer_FkCt · 2024-11-25
> > **Acknowledgement**
> >
> > Thank you for the thoughtful response, I have raised my score.

---

> > > ### Author Response · Authors · 2024-12-01
> > > **Reply**
> > >
> > > Thanks for your positive response to our rebuttal.
> > >
> > > As a question from our side, we are interested in hearing why you find our work only of "fair" contribution? Hearing your thought on this would help us improve our manuscript further, as we want our manuscript to communicate to a broad range of readers. If you have any additional suggestions/concerns, we are happy to answer and adapt our final version accordingly.

---

### Official Review · Reviewer_vsHn · 2024-11-03

**Soundness:** 3
**Presentation:** 3
**Contribution:** 3
**Rating:** 8
**Confidence:** 3

**Summary:**

The paper introduces six alignment relations to quantify and analyze model representations, collectively termed the Canonical Representation Hypothesis (CRH). This hypothesis proposes that neural networks naturally develop compact representations, with neurons and weights remaining unchanged by transformations that do not affect the task. The study theoretically examines conditions under which CRH holds and demonstrates that when CRH is disrupted, reciprocal power-law relationships arise among representations, weights, and gradients—a phenomenon referred to as the Polynomial Alignment Hypothesis (PAH). Together, CRH and PAH provide a potential unified theory that encompasses several significant phenomena in representation learning, such as Neural Collapse and the Neural Feature Ansatz.

**Strengths:**

First of all, congratulations on your ICLR submission! I’d like to commend the author for this great piece of research. Below is my review of the work:

**Originality**
This paper introduces a unifying framework for studying neural network representations, with a notable emphasis on backward representation—a relatively unexplored area.

**Quality**
The paper is of generally high quality:
- It acknowledges its own limitations effectively.
- Experimental results are detailed and well-aligned with theoretical insights.
- Findings show that the theory holds across different types of networks.
- Two Mechanisms that disrupt alignment are identified. Quantification is provided via the Polynomial Alignment Hypothesis.

**Clarity**
The paper is clearly written in good English and is well-structured. Figures are clear, informative, and effectively support the content.

**Significance**
This paper offers a potentially unifying framework for studying neural network representations and identifying key alignment metrics that inform network alignment. It contributes to an active research area and presents predictions relevant to widely used models like ResNet and LLMs, enhancing its impact in the field. Furthermore, this research could have potential interdisciplinary applications such as neurosciences.

**Weaknesses:**

**Originality**
Further clarifying how this work uniquely contributes to and advances the field would enhance its perceived originality.

**Quality**
No clear weakness in the quality.

**Clarity**
-  The term "hypothesis" may be somewhat misleading, as the proposed concept functions more as a measurement framework rather than a traditional hypothesis. Reframing it as such would emphasize its role in quantifying neural network representations without introducing potentially confusing terminology.
- The organization of the paper could benefit from refinement. Early references to later sections (such as mentioning Figure 9 before the relevant context is provided) disrupt the flow, and frequent shifts in reference to the later text and figures make the argument more challenging to follow.
- Providing more detailed explanations for terms such as "neural collapse" and "alignment" would enhance the paper's accessibility.
- Figure 4 lacks a colour bar for the weight decay parameter, which complicates the interpretation of results.

**Significance**
To improve the paper’s significance, it would be helpful to expand on the implications of the different alignment phases. For instance, as mentioned in the text, positive exponents between Za and Ha suggest that the layer enhances the principal components of Ha while diminishing lesser features, whereas a negative exponent indicates the opposite. Greater detail on these implications would improve interpretability, especially regarding how these phases contribute to understanding network behaviour. Additionally, exploring factors that lead to alignment breakdown would deepen the paper's contribution to interpretability in this area.

**Questions:**

- How might different initialization strategies impact the different observed alignment phenomena?
- Could the authors elaborate on how the phases correspond to observable behavioural or performance changes in the model?
- Beyond the two mechanisms identified in the paper, could other factors contribute to the disruption of alignment?
- How might the different alignments discussed in this paper relate to concepts from the rich and lazy learning literature? My understanding is that the neural feature ansatz describes and quantifies the rich regime by analyzing feature evolution. Could these alignments and phases potentially provide a way to further describe and quantify distinct learning regimes ( Specifically further define the different types of rich regimes)?

I hope this review will be helpful in the further development of the paper. I encourage the author to continue this research and incorporate the feedback provided. With improvements in structure, clarification of key concepts, and development of the practical significance of these hypotheses, this paper has the potential to earn a higher score.

---

> ### Author Response · Authors · 2024-11-20
> **Rebuttal Part 1**
>
> Thanks for your positive and detailed feedback. We answer both the weaknesses and questions below.
>
> **"I’d like to commend the author for this great piece of research."**
>
> Thanks for this encouragement. While we are glad that you find our work a "great piece of research," we feel rather confused that a great piece of research only receives a score of "marginally below the acceptance threshold."
>
> **Weaknesses**
>
> **Originality Further clarifying how this work uniquely contributes to and advances the field would enhance its perceived originality.**
>
> Thanks for giving us the chance to answer this important question, which we only concisely touched on in the manuscript due to space limitations. Please see the **Relevance and Implication** part of the summary rebuttal for our answer to this question. We believe that the study of these alignment relations can have broad and interdisciplinary implications for (1) theory and science of deep learning, (2) practice of deep learning, (3) physics, and (4) neuroscience.
>
> **Clarity. The term "hypothesis" may be somewhat misleading, as the proposed concept functions more as a measurement framework rather than a traditional hypothesis. Reframing it as such would emphasize its role in quantifying neural network representations without introducing potentially confusing terminology.**
>
> Thanks for this criticism. While we are happy to change the word “hypothesis” if there is a better alternative, please let us explain why the CRH is both a framework and a hypothesis (in a way, it is a framework of multiple hypotheses). The word "hypothesis" is defined to mean that "a tentative assumption made in order to draw out and test its logical or empirical consequences" (https://www.merriam-webster.com/dictionary/hypothesis).
>
> The CRH ("that all six alignment conditions hold after training") is clearly a tentative assumption, whose empirical consequences are to be tested. It therefore matches the common understanding of what it means to be a hypothesis. The sequence of measurement procedures that one carries out in order to test whether the CRH is broken, and whether the model belongs to the PAH, is, certainly, a measurement framework. Therefore, one can without confusion speak of both the "CRH framework" and the "CRH hypothesis" to mean related but subtly different things.
>
>
>
>
> **The organization of the paper could benefit from refinement. Early references to later sections (such as mentioning Figure 9 before the relevant context is provided) disrupt the flow, and frequent shifts in reference to the later text and figures make the argument more challenging to follow.**
>
> Thanks for this criticism. We have clarified some of the references and will carefully adapt the main text to avoid references that may interrupt the reading before our final version.
>
> **Providing more detailed explanations for terms such as "neural collapse" and "alignment" would enhance the paper's accessibility.**
>
> Thanks for this suggestion. Neural collapse is a well-known phenomenon in the study of the theory of deep learning. A main implication of neural collapse is that the inner class variations in the representations vanish, and so the representation is robust to, for example, adversarial attacks (see Figure 8 in arxiv.org/abs/2008.08186). We have added a little more discussion of this in Section A. Also, in our work, alignment refers to the relationship between matrices A and B such that the eigenvectors or the eigenvalues of A (or both) become the same as that of B. We will clarify these in our final version.
>
>
>
> **Figure 4 lacks a colour bar for the weight decay parameter, which complicates the interpretation of results.**
>
> Thanks for this question. We now include a full version of Figure 4 in Figure 22 in Section F to show the weight decay values for each point. The reason why we do not show this in the main text is that the figure with full details is a little too messy.

---

> ### Author Response · Authors · 2024-11-20
> **Rebuttal Part 2**
>
> **To improve the paper’s significance, it would be helpful to expand on the implications of the different alignment phases. For instance, as mentioned in the text, positive exponents between Za and Ha suggest that the layer enhances the principal components of Ha while diminishing lesser features, whereas a negative exponent indicates the opposite. Greater detail on these implications would improve interpretability, especially regarding how these phases contribute to understanding network behaviour. Additionally, exploring factors that lead to alignment breakdown would deepen the paper's contribution to interpretability in this area.**
>
> Thanks for this suggestion. These are indeed important future problems. Due to the hard space limit of ICLR (10 pages), we cannot include more detailed discussions in the main text. To help the readers better understand these parts, we included a new section in Appendix D to discuss  how a phase of the PAH is directly related to the solution of simple linear regression, which we believe to help the readers better understand these effects. In this example, the linear regression can be interpreted to have applied a normalization of the input distribution, which leads to $H_a^{-1} \propto Z_a$. This example shows that the PAH may be identified implicitly even in well-known examples.
>
> Exploring factors that lead to alignment breakdown is indeed an important problem, which we have studied in Figure 3, 7, 8. On the theory side, we have suggested three mechanisms for its breaking, supported with either theory or experiment. We believe this is sufficient for the claims we are making. One should regard our present work as the first step towards building a unified framework towards understanding the formation of representation in neural networks. More explorations would be valuable but we kindly believe that they would be out of the scope of our current manuscript.
>
>
> **Questions:**
>
> **How might different initialization strategies impact the different observed alignment phenomena?**
>
> This is an open but important future step. A limitation of our present theory is that it only deals with the final stage of training. An important future step is to study the learning dynamics of these phases, which should reveal how each phase is related to the initialization. However, at this moment, this question remains an open problem. Also, see our answer below regarding feature learning, as initialization schemes are closely related to whether a network is in the lazy training or the feature learning regime.
>
> **Could the authors elaborate on how the phases correspond to observable behavioural or performance changes in the model?**
>
> This question is an important open problem. Intuitively, it does seem very likely that these phases will imply different learning dynamics and modes of information processing in the network. The answers we give here are only educated guesses of what might happen -- at this moment it is difficult to check whether these statements are true or not because we do not yet have any systematic method for inducing a specific phase, which is an important future problem.
>
> One way the phase could imply we suggested in the manuscript is that a positive relation between $H_a$ and $Z_a$ could imply a low-rank mechanism, where larger eigenvalues of $H_a$ are expanded while smaller eigenvalues are suppressed. That neural networks learn these low-rank representations is consistent with common observation. For example, see https://arxiv.org/abs/2410.23819
> Another implied mechanism is representation normalization. This happens when $H_a \propto Z_a^{-1}$, which means that $H_b$ will be normalized after this weight. See the newly added discussion about linear regression in Section D for how this phase exists implicitly even in linear regression.
>
> Another kind of interesting phase is where the alignment condition implies gradient vanishing or explosion problems, which have been well-known for decades. Noticing that $g_a =W^T g_b$, it is naturally the case that if $Z_b$ and $G_b$ are aligned with a positive exponent, the gradient in the previous layer will face an explosion and vanishing problem simultaneously (as larger eigenvalues becomes larger). In contrast, the phase $Z_b\prop G_b^{-1}$ seems to be the ideal training phase as it implies that the gradient in the previous will be normalized. Identifying causes of these phases are also of great future interest.
>
> We have added the above discussion to the manuscript.
>
>
> **Beyond the two mechanisms identified in the paper, could other factors contribute to the disruption of alignment?**
>
> We believe there are. One known mechanism is that the directions of the activations/gradients show systematic dependence on their norms. This point is demonstrated in and discussed briefly in the text around Figure 7. We believe that an important future step is indeed to find more causes to break the CRH.

---

> ### Author Response · Authors · 2024-11-20
> **Rebuttal Part 3**
>
> **How might the different alignments discussed in this paper relate to concepts from the rich and lazy learning literature? My understanding is that the neural feature ansatz describes and quantifies the rich regime by analyzing feature evolution. Could these alignments and phases potentially provide a way to further describe and quantify distinct learning regimes ( Specifically further define the different types of rich regimes)?**
>
> This is a great question. They are indeed closely related. Essentially, all the alignments of the CRH should happen in the feature learning regime. This is because in lazy training, the change in the weights are negligible and so, for example, the weight outer products are the same as initialization. Likewise, the representations are also the same as the initialization and because everything is random there should be weak or no alignments -- and the alignments should not change during training. One can also see this from the discussion in the standard feature learning literature -- for example, that the weights and representation becomes correlated is a major mechanism for feature learning to happen (see arxiv.org/abs/2311.02076).
>
> And yes, all the subphases of the PAH belong to the feature learning regime and are indeed good ways to "classify" or categorize feature learning into different regimes. In this sense, one important contribution (which we did not discuss in detail in the manuscript due to space constraint) is a fine-grained categorization of different feature learning regimes. This may be used by the feature learning community to better understand the scaling limits of neural networks.

---

> > ### Comment · Reviewer_vsHn · 2024-11-24
> >
> > I found the study to be interesting and of good quality.  However, as noted in my initial review, the paper required improvements in its structure, clearer articulation of key concepts, and more thorough development of the practical implications of its hypotheses. The lack of clarity, in particular, influenced my initial scoring. Additionally, I felt there was a few weaknesses and questions that needed further explanation. I hope this clarifies any confusion regarding my previous evaluation while also acknowledging the strengths of this research.
> >
> > In the rebuttal, I commend the authors for their efforts in providing clear explanations and detailed responses to the questions raised. I found the explanation of the work's relevance, the clear connection to the rich and lazy literature (as a categorization of different feature learning regimes), and the implications of alignment, particularly insightful.
> > I strongly recommend avoiding including a figure without a legend in the main text, even if a detailed version is provided in the appendix ( or only put the figure in the appendix altogether).
> >
> > I hope my feedback proves helpful.
> > Taking all considerations into account, I am increasing my overall score to 8.

---

> > > ### Author Response · Authors · 2024-12-01
> > > **Reply**
> > >
> > > Thank you for carefully explaining your thoughts. We will fix up Figure 4 and check through the main text thoroughly to improve its clarity and emphasize our contribution further in our final revision.

---

### Official Review · Reviewer_84cQ · 2024-11-04

**Soundness:** 4
**Presentation:** 4
**Contribution:** 3
**Rating:** 8
**Confidence:** 2

**Summary:**

The authors propose an overarching theory for neural network learning which encompasses the alignment of representations, gradients, and weights throughout training. This framework uniquely combines a diverse set past work under a unified 'canonical representation hypothesis' (CRH) umbrella, and prove theoretically that the CRH does hold. Succinctly, they show that gradient nose 'expands the representation' and weight decay 'contracts it', and they must reach stationarity by balancing these terms. They then use this framework to make predictions about neural network training when each of the six elements of the CRH is broken, provided a set of 64 potential phases for neural network training.

In conclusion,the paper is well written, organized, and situated in the literature. While I am not an expert in this domain, I therefore defer to other more informed reviewers to make statements about the novelty and accuracy of the authors' claims. Otherwise, the results appear impressive and impactful for the theoretical neural network community, and thus carry significant value.

**Strengths:**

- The paper is well written, and carefully presented to ensure correctness and clarity.
- The framework appears general, encompassing significant past work.
- The theorems are general, avoiding reference to any specific loss function or activation functions, helping to explain their ubiquity.
- The experimental results appear to strongly support the developed theory and appear both rigorous and extensive.
- The insights section is very much welcomed, and provides significant value to readers coming from outside the primary domain.
- The limitations are clearly stated.

**Weaknesses:**

- The authors state that the CRH holds better for later layers than earlier layers.

**Questions:**

- Can you explain the limitation of the theory exposed by Figure 7? Why do you believe that you see the three branches as opposed to one?

---

> ### Author Response · Authors · 2024-11-20
> **Rebuttal**
>
> Thanks for the detailed feedback! We answer both the weaknesses and questions below.
>
> **The authors state that the CRH holds better for later layers than earlier layers.**
>
> Thanks for this comment. This is somewhat expected because, as the theory (Theorem 3) suggests, the CRH is a generalization of neural collapse, which is found to occur in the later layers of a neural network. This observation is thus consistent with the previous literature. In this sense, that the CRH holds better for later layers is not a weakness but a feature of the theory. At the same time, a lot of our results are devoted to understanding the breaking of CRH; for example, the PAH (or other subsets of the CRH) is found to hold quite well for the earlier layers in experiments. For example, this is clear both from the experiments in Section C6 and from the supplementary code we provided.
>
> **Can you explain the limitation of the theory exposed by Figure 7? Why do you believe that you see the three branches as opposed to one?**
>
> Thanks for raising this question. This limitation has been briefly discussed in lines 439-441. A major assumption in the theory is that the direction of the gradients/activations is noncorrelated to its norm (assumption 1). That there are three branches is a sign that there is indeed some systematic dependence of the gradient/activation direction on its norm. The reason or any characterization for this dependence is yet unknown and could be an interesting future problem to study for a better understanding of modern deep learning.

---

> > ### Comment · Reviewer_84cQ · 2024-11-25
> >
> > We thank the authors for precisely addressing our questions. The additional relevance and implications are also greatly appreciated, and the relation to biological neural representations is quite exciting. I find the updates to the paper to be helpful and I will retain my high rating of the paper. I would additionally be a proponent of raising the score in discussion with the other reviewers since I find the work quite insightful and it appears there are many new research directions being illuminated from this analysis. I have no further concerns.

---

> > > ### Author Response · Authors · 2024-12-01
> > > **Reply**
> > >
> > > Thank you for your constructive feedback and for advocating our work. We do plan to explore the biological implications more in the future, and our current work will be an important step towards that.

---

### Official Review · Reviewer_LWzQ · 2024-11-08

**Soundness:** 3
**Presentation:** 3
**Contribution:** 4
**Rating:** 8
**Confidence:** 3

**Summary:**

First of all, I apologize for the late review with the Authors, the AC, Senior AC and the PC.

This paper introduce the Canonical Representation Hypothesis (CRH), positing that in neural networks, as an effect of training, the representations, weights, and gradients (with respect to the representations) within a layer align with each other.

This alignment is seen as an essential part of how neural networks form structured, invariant representations.
The Authors introduce the CRH and support it with a theory that shows how this alignment emerge as a consequence of stationarity towards the end of training, through a balance of gradient noise and regularization.

Additionally, they propose the Polynomial Alignment Hypothesis (PAH), which describes patterns in cases where CRH is broken, leading to specific scaling relationships in neural networks, of a different nature of scaling laws involving network performances.

This framework offers a new perspective on understanding important aspects of neural representations, unifying phenomena such as neural collapse and feature learning.

**Strengths:**

The CRH provides a new framework to interpret basic statistical aspects of how neural networks develop representations. The system of CRH equations generalize and organize into a coherent view previous, partial attempts to reveal alignment phenomena between representations, gradients and weights.  By relating CRH to neural collapse, the Authors offer a broader view that positions this phenomenon as a specific case of a more universal behavior in neural networks.

The framework is very general, focusing on alignment rather than on specific network designs or loss functions. This  increases the potential impact of CRH on future studies, as it provides a general model for understanding basilar statistical aspects of representation formation.

The Authors present a detailed theoretical derivation supporting the CRH. The theoretical derivations adds credibility to their claims, encouraging further studies, in particular related to dynamics.

**Weaknesses:**

While this work aims at universality, the Authors are the first to recognize that their focus is on specific types of networks and tasks, commenting that further empirical tests across more diverse datasets and architectures are necessary to confirm the CRH and PAH in practice.

The weakness I found is that certain assumptions - which are critical for the theoretical foundation of CRH - may not hold in general, like the self-averaging conditions (Hypothesis A.1, A.2). I think the work woud benefit for an empirical assessment on how severe is the departure from self-averaging in a concrete, relevant case.

Furthermore, I found particularly difficult to follow Section B (THEORY AND PROOFS) of the Supplementary Materials. For example, I could not figure out, Lemma 1 in Section B, mainly because some of the "actors" in the scene (like the A tilde in eq. 21) were not presented first. But it may be also a lack of understanding from my side.

However, adding a few more explanations in this section would improve readibility.

**Questions:**

Please refer to the weakness section.

**Details Of Ethics Concerns:**

No concerns.

---

> ### Author Response · Authors · 2024-11-20
> **Rebuttal**
>
> Thanks for the detailed feedback! We answer both the weaknesses and questions below.
>
> **While this work aims at universality, the Authors are the first to recognize that their focus is on specific types of networks and tasks, commenting that further empirical tests across more diverse datasets and architectures are necessary to confirm the CRH and PAH in practice.**
>
> Thanks for pointing this out. We believe and agree that an important future direction of empirical work is to verify or falsify our CRH framework across diverse training settings, and these empirical tests will better our understanding of modern neural networks.
>
> **The weakness I found is that certain assumptions - which are critical for the theoretical foundation of CRH - may not hold in general, like the self-averaging conditions (Hypothesis A.1, A.2). I think the work woud benefit for an empirical assessment on how severe is the departure from self-averaging in a concrete, relevant case.**
>
> Thanks for this comment. We actually have an experiment in the Appendix testing the stationarity of these matrices in ResNet18. See Figure 21 in Section C9. We see that at the end of the training, the representation covariance reaches stationarity in all layers of ResNet18 trained on CIFAR10. An additional intriguing observation from this figure is that the convolutional layers reach stationarity as well -- this could imply that some forms of CRH (after taking the weight-sharing effect into account) might apply to the convolutional layers as well, and this could be an interesting future step.
>
> **Furthermore, I found particularly difficult to follow Section B (THEORY AND PROOFS) of the Supplementary Materials. For example, I could not figure out, Lemma 1 in Section B, mainly because some of the "actors" in the scene (like the A tilde in eq. 21) were not presented first. But it may be also a lack of understanding from my side. However, adding a few more explanations in this section would improve readibility.**
>
> Thanks for asking for clarification. Lemma 1,2,3 are generic statements about arbitrary symmetric matrices $A$ and $B$, which is clear from the lemma statement. Eventually, when we apply these lemmas, the specific $A$ and $B$ will be the H, G, and Z matrices we defined in the CRH (Eq. 3-5). We have added an explanation for the purpose of these lemmas in Section B to help with understanding.

---

> > ### Comment · Reviewer_LWzQ · 2024-11-27
> > **Reply to the Authors**
> >
> > I thank the Authors and the Reviewers for clarifying on aspects that I did not raise. I appreciate more the scope of the paper, and the promised effort to clarify the sections I found difficult to interpret. Therefore, I confirm my grade! Thanks again also the the AC for the kind conduction of the process.

---

> > > ### Author Response · Authors · 2024-12-01
> > > **Reply**
> > >
> > > Thanks for your reply. We are encouraged to hear that our rebuttal has addressed your concerns. We will carefully proofread the theory section again in the final version to improve its clarity and avoid confusion.

---

> ### Comment · Area_Chair_HNo2 · 2024-11-27
> **Rebuttal Response**
>
> While the review was quite positive, it would still be appreciated if you could respond to the rebuttal and note if the other reviews had any impact on your review. Thanks!

---

### Author Response · Authors · 2024-11-20
**Summary Rebuttal**

We thank all the reviewers for the careful and constructive feedback. We are also encouraged to hear that all reviewers find our results interesting and relevant. For examples, reviewer LWzQ08 rates our contribution as "excellent", and reviewer 84cQ rates our soundness and presentation as "excellent."

To address the criticisms from the reviewers, we made the following additions to the draft (colored in orange). We believe we have addressed the concerns of the reviewers. We invite all the reviewers to ask additional questions, and we would be very happy to address them. We also give more detailed explanations to each reviewer, respectively.

1. An illustration and explanation of one of the learning phases of the PAH in an ordinary and simple example (linear regression) in section D.

2. Discussion of how different phases of the PAH may be of interest for future studies and their relationships with prior well-known results or observations in the field, in section E.

3. The full figure for Figure 4 is shown in Section F

Many other changes to the main text are also made to fix typos and avoid potential confusion. In the next part, we discuss the broad interdisciplinary relevance and implications of our results.

---

> ### Author Response · Authors · 2024-11-20
> **Summary Rebuttal (part 2)**
>
> **Relevance and Implications**. A major question from reviewer vsHn and FkCt is about how the CRH we discovered may help advance the field. While we discussed these relevances in more concise forms in the main text, we would like to make an extended discussion here to better clarify the importance and relevance of our results. We believe that our results are of an interdisciplinary nature, and fully understanding them may require knowledge from different fields, which we discuss below: (1) theory and science of deep learning, (2) practice of deep learning, (3) physics, and (4) neuroscience.
>
> 1. Understanding deep learning. The most important contribution is certainly a mechanism and a hypothesis framework for understanding representation learning in neural networks. Besides its direct relationship to neural feature ansatz and neural collapse, our result is also related to the information bottleneck theory (arxiv.org/abs/1503.02406), which hypothesizes that neural networks learn a compact representation after training. Our theory and empirical results show one exact mechanism that the information bottleneck hypothesis might be true. The neural collapse phenomenon has also been of great interest in the theory community and found to be relevant to transfer learning (arxiv.org/abs/2112.15121) and imbalance classification (arxiv.org/abs/2203.09081), and our result extends the neural collapse formalism to broader settings such as generic regression and self-supervised learning.
>
> - Our result also sheds light on how feature learning happens in neural networks. As we briefly discussed in the main text, these alignments are signs of feature learning because they do not arise in the NTK/lazy regime. This is because in lazy training, the change in the weights is negligible, and so, for example, the weight outer products are the same as initialization. Likewise, the representations are also the same as the initialization, and because everything is random, there should be weak or no alignments -- and the alignments should not change during training.
>
> - One can also see this from the discussion in the standard feature learning literature -- for example, that the weights and representation become correlated is a major mechanism for feature learning to happen (see arxiv.org/abs/2311.02076). This implies that all the subphases of the PAH belong to the feature learning regime and are indeed good ways to "classify" or categorize feature learning into different regimes. In this sense, one important contribution (which we did not discuss in detail in the manuscript due to space constraints) is a fine-grained categorization of different feature learning regimes. This may be used by the feature learning community to better understand the scaling limits of neural networks.
>
> 2. Potential Inspirations for neuroscience:  Understanding how latent representations are formed in animal brains is a fundamental scientific problem in neuroscience. Until now, there has been a very limited understanding of this, and our theory suggests one mechanism of how latent representations form and can inspire neuroscientists to understand biological brains. The fact that noise and regularization balance the solutions can be a core mechanism for latent representation formation is a novel and potentially surprising message. For example, compare the latent representations in our paper with the biological data in Figure 7B in www.cell.com/neuron/fulltext/S0896-6273(18)30581-6
>
> 3. Bridging Physics and deep learning: Using insights and concepts from physics and science to study modern artificial neural networks is an emergent field. Our theoretical part shows that the fluctuation dissipation theorem, one of the fundamental laws of thermodynamics, may be a great relevance in understanding how learning happens in neural networks. This can inspire further developments in the science of deep learning and lead to cross-fertilization of both the fields of AI and physics.
>
> 4. Practice of deep learning:  Lastly, our theory also has implications for the practices of deep learning, although we do not claim this to be a main contribution. As one such example, a lot of works are devoted to shaping the representations of neural networks to match certain structures (e.g., see arxiv.org/abs/1707.07847). Assuming RGA (representation-gradient alignment), our work suggests that one can engineer the learned representation by introducing noises to the neurons (in this sense, dropout can be interpreted as performing "representation engineering"). Therefore, our theory implies novel algorithms for feature/representation engineering.

---

### Meta-Review · Area_Chair_HNo2 · 2024-12-17

**Metareview:**

**Summary** This paper proposes the Canonical Representation Hypothesis (CRH), a hypothesis expressed in terms of six relations that after training the representations, weights and gradients align in each layer. The alignment is proposed as a fundamental part of understanding how neural networks form compact invariant representations.  The paper also proposes the Polynomial Alignment Hypothesis (PAH) which describes cases where CRH is broken.

**Strengths** Reviewers were excited that the paper offered a new framework for understanding representation learning.  In particular, CRH seems to unify and generalize several previous efforts in the literature, including explaining neural collapse and  neural feature ansatz. The framework is quite general, applying broadly to different architectures and datasets.  Reviewers also praised the theoretical derivation supporting CRH and noted the experimental results supported the theory.

**Weaknesses** Few weaknesses were noted.  Initially there were questions about the paper clarity and significance, but these were addressed during the author-reviewer discussion.  It was also noted the theoretical results rely on assumptions which may not hold in practice.

**Conclusion** Reviewers were unanimously in favor of acceptance and generally praised the paper as a significant contribution to the theory of representation learning in neural networks.

**Additional Comments On Reviewer Discussion:**

Reviewer vsHn initially gave a 5 based on weaknesses in paper structure clarify and significance but increase their score to 8 after author discussion.  Reviewer FkCt also had questions about significance and motivation and increased their score from 5 to 6 after the author response.

---

### Decision · Program_Chairs · 2025-01-22

Accept (Spotlight)